# Origin, transport, and retention of fluvial sedimentary organic matter in South Africa's largest freshwater wetland, Mkhuze Wetland System

Julia Gensel[1], Marc Steven Humphries[2], Matthias Zabel[1], David Sebag[3,4], Annette Hahn[1], and Enno Schefuß[1]

[1]MARUM - Center for Marine Environmental Sciences, University of Bremen, Bremen, Germany
[2]School of Chemistry, University of the Witwatersrand, Johannesburg, South Africa
[3]IFP Energies Nouvelles, Direction Sciences de la Terre et Technologies de l'Environnement, France
[4]Normandie Univ, UNIROUEN, UNICAEN, CNRS, M2C, 76000 Rouen, France

**Correspondence:** Julia Gensel (julia.gensel@mail.de)

**Abstract.** Sedimentary organic matter (OM) analyses along a 130 km-long transect of the Mkhuze River from the Lebombo Mountains to its outlet into Lake St. Lucia, Africa's most extensive estuarine system, revealed the present active trapping function of a terminal freshwater wetland. Combining bulk OM analyses, such as Rock-Eval, and source-specific biomarker analyses of plant-wax $n$-alkanes and their stable carbon ($\delta^{13}$C) and hydrogen ($\delta$D) isotopic composition showed that fluvial sedimentary OM originating from inland areas is mainly deposited in the floodplain and swamp area of the wetland system but not in the downstream lake area. A distinctly less degraded OM signature, i.e., considerably higher I-index and lower R-index as well as recognizable higher $\delta$D values compared to samples from upstream sub-environments characterize surface sediments of Lake St. Lucia. The offset in $\delta$D indicates that the contributing vegetation, although similar to upstream vegetation inputs in terms of photosynthetic pathway ($\delta^{13}$C) and alkane distribution pattern, experienced different hydrological growth conditions. The results suggest that under current conditions hinterland sedimentary OM is deposited throughout the wetland system up to the Mkhuze Swamps, which ultimately captures the transported OM. Consequently, samples from the downstream located Lake St. Lucia show locally derived signals instead of integrated signals encompassing the river catchment. This finding raises important constraints for future environmental studies as the assumption of watershed-integrated signals in sedimentary archives retrieved from downstream lakes or offshore might not hold true in certain settings.

## 1 Introduction

Lake St. Lucia is the largest estuarine system in Africa and is of both local and international importance (Porter, 2013). It is a biodiversity hotspot and provides habitat for fish, wildlife, and breeding grounds for birds. On this basis, it significantly supports national and international ecotourism as well as the regional economy (Whitfield and Taylor, 2009). Its functioning is highly dependent on a constant supply of freshwater. One of the major hydrological inputs into Lake St. Lucia is the Mkhuze Wetland System, South Africa's largest freshwater wetland system, which accounts for about 56 % of the freshwater input to the lake (Stormanns, 1987). It is part of the iSimangaliso Wetland Park, which was declared a UNESCO World Heritage Site in

1999 (Taylor et al., 2006), was designated a Ramsar site in 1986 (Perissinotto et al., 2013), and is the World's oldest formally protected estuary (Whitfield and Taylor, 2009).

Wetlands act as filters by removing or retaining nutrients and man-made pollutants (Reddy and Debusk, 1987). They also control sedimentation through deposition, resulting in less siltation in adjacent water systems (Johnston, 1991). The filtering function of wetlands is of particular interest because wetlands are considered to be either sources or sinks of organic carbon and therefore may increase carbon emissions from surface waters if they export organic carbon (OC) in substantial amounts to adjacent water systems and vice versa (e.g, Cole et al., 2007; Reddy and DeLaune, 2008; Mitsch and Gosselink, 2015). Factors controlling whether a wetland serves as a carbon source or sink are not yet fully understood, but wetland hydrology is considered to play a critical role. When a wetland's natural filter function is overstressed, the wetland and its beneficial functions can be reduced or destroyed (Hemond and Benoit, 1988; Junk and An, 2013). In the worst case, the wetland can become a source of pollutants and sediments that have been previously stored. The Mkhuze Wetland System, and in particular the Mkhuze Swamps, is supposedly such a filter for Lake St. Lucia's water supply and therefore presumably provide the benefits mentioned above.

Like most wetlands, the Mkhuze Wetland System is affected by human interference. For instance, cattle grazing and use as agricultural land are reported to be of increasing importance for both sugarcane and eucalyptus cultivation (Neal, 2001). Human activities such as channel dredging (Mpempe and Tshanetshe canals) have also significantly altered the natural state (Neal, 2001; Barnes et al., 2002; Ellery et al., 2003). Such disturbances have the potential to cause alteration of vegetation, hydrologic conditions, as well as sediment balance and transport pathways. Such negative impacts have extensively been studied for the swamp system formerly present to the south of Lake St. Lucia, the Mfolozi Swamps (Whitfield and Taylor, 2009; Taylor, 2013, and references therein).

Our study is, to the best of our knowledge, the first to examine the sources, transport and fate of organic material in the Mkhuze Wetland System. The primary objective of the study are (i) to characterize the sedimentary OM in the individual sub-areas of the system and (ii) to infer OM sources and transport as well as degradation of OM within the system. We, therefore, assume that rivers transport sedimentary material (if so) only downstream. In this context, signals observed in downstream areas can only originate from upstream areas through transport or from local sources. Thereby, we evaluate the current wetland's impact, especially of the Mkhuze Swamps on Lake St. Lucia. To adress the wetland's role in retaining sedimentary OM, we analyzed surface sediments along a 130 km-long transect from the Mkhuze River channel to its mouth in Lake St. Lucia, as well as extending into the northern part of Lake St. Lucia.

The provided insights into the characteristics and transport of organic matter in a terminal wetland under subtropical climatic conditions, which also can be identified in other wetlands around the World reveal potential implications for (pa-leo)environmental studies based on sedimentary archives from downstream areas.

## 2  Approach and methodological background

To obtain qualitative information on OM characteristics, we applied a combination of methods. Bulk methods (bulk organic and Rock-Eval analyses) characterize the entire OM and its degradation state but are limited in their specificity, whereas compound-specific analyses, such as the identification of lipid biomarkers (i.e., plant-wax derived *n*-alkanes) and their compound-specific isotope composition provide very specific information but reflect only a small proportion of OM.

### 2.1  Rock-Eval

Originally developed to provide information on the content of hydrocarbons and the kerogen type in sedimentary rocks, as well as determine the maturity of the kerogen (Disnar, 1994; Lafargue et al., 1998), Rock-Eval analysis has become an attractive and fast alternative for the analysis of organic matter in soils and sediments compared to conventional methods due to the lack of sample preparation (Disnar et al., 2003; Sebag et al., 2006). Rock-Eval analysis provides not only the quantity of organic carbon but also information on the elemental composition of the OM. In particular, the Hydrogen Index (HI; mg HC/g TOC) and Oxygen Index (OI; mg $O_2$/g TOC), which correlate with the classical elemental parameters of the H/C and O/C ratios known from Van Krevelen diagrams (Espitalié et al., 1977, 1985) provide insights into the degree of degradation, but also provide indications on OM origin. The representation of the HI against TOC allows the estimation of the soil OM quality, which is usually determined by the mineralization of the soil OM and the dilution by the mineral matrix. Fresh plant OM and litter usually show high TOC levels (10-40 %) and HI values greater than 300 (Disnar et al., 2003). This results from the material consisting primarily of unconverted biopolymers such as lignin, cellulose, and hemicellulose. During degradation (oxidation) of the biopolymers, the HI decreases and the OI increases. Material consisting of a distinct composition of the dominant biopolymers, such as aquatic inputs, likewise, shows distinct ranges of its HI values which provide a rough source indication.

In addition, Rock-Eval provides information on the thermal stability of the immature OM (Disnar, 1982). The so-called S2 peak integrates the quantity of pyrolyzed emission products between 200 °C and 650 °C. Utilizing mathematical deconvolution determines individual peaks within the S2 peak. Fractions of OM differing in thermal stability are assignable to the resulting temperature intervals: (A1) 200 °C - 340 °C: thermally unstable biological macromolecules, like saccharides, (A2) 340 °C - 400 °C: stable biological macromolecules, like cellulose and/or lignin or polypeptides, (A3) 400 °C - 460 °C: immature geological macromolecules, like humic substances, (A4) 460 °C - 520,°C: refractory geological macromolecules, and (A5) 520 °C - 650,°C: highly refractory pool (Disnar et al., 2003; Sebag et al., 2006, 2016). [13]C NMR investigations of composts showed that the thermal stability separating the mentioned groups also correlates with the chemical stability (Albrecht et al., 2015). Based on this, Sebag et al. (2006) and Albrecht et al. (2015) developed indices based on the aforementioned temperature intervals, whereby the indices are defined as follows: R-index (R=recalcitrant OM) = (A3 + A4 + A5)/100 and I-index (I=immature OM) = log10((A1 + A2)/A3). However, Sebag et al. (2016) points out that due to their mathematical derivation, these indices cannot exactly indicate the quantity or identity of the chemical components in the OM, unlike the classic Rock-Eval parameters. We acknowledge that the four different temperature intervals cannot differentiate the entire spectrum of

organic compounds and therefore refer to the thermal rather than chemical stability of OM (Sebag et al., 2016). Nevertheless, this sufficiently reflects the changes in the organic components during their transformation. On this basis, the R-index and the I-index describe the contribution of refractory and persistent fractions by pedogenic processes, as well as the degree of transformation of unstable components via decompositional processes (Sebag et al., 2016). Sebag et al. (2016) proposed the use of the I/R diagram (I-index vs. R-index), which resembles the representation of the Van-Krevelen diagram (OI vs. HI). One of the most notable features of the I/R diagram is the linear relationship between the two parameters ("decomposition regression line" or "humic trend"). During the decomposition of fresh OM, labile organic compounds are converted into their respective more stable counterparts, which are either converted into stable humic acids or components complexed by the mineral matrix. Samples plot along the linear relationship when the stabilization of OM from the progressive degradation of organic components is based on their biochemical stability. Samples do not plot along this linear relationship in situations with OM mixture from different sources.

## 2.2 Plant waxes

In general, lipid biomarkers reflect only a small portion of the total OM but overcome problems of source specificity by having a specific origin. Plants, for instance, produce waxes to control their water balance and counteract external stressors, such as UV radiation, bacterial and fungus-based attacks (Jenks and Ashworth, 1999). These waxes enter the environment through physical processes, such as wind, precipitation, abrasion by particles and when plants die or plant parts detach. The waxes contain a variety of molecules, including long-chain *n*-alkanes, which are very resistant to degradation (Poynter and Eglinton, 1991). Plant-wax-derived *n*-alkanes show homologous series with a strong odd-over-even chain-length dominance due to their biosynthesis (Eglinton and Hamilton, 1967) causing a high carbon preference index (CPI) for fresh plant waxes. Besides, different plant types show characteristic features in their *n*-alkane distribution patterns, such as aquatic plants showing dominant contributions of the *n*-alkanes $C_{23}$ and $C_{25}$ (Ficken et al., 2000; Baas et al., 2000; Liu et al., 2019), whereas grasses show distribution patterns dominated by the very long-chain *n*-alkanes $C_{33}$ and $C_{35}$ (Rommerskirchen et al., 2006; Vogts et al., 2012). The homologue $C_{29}$ seems to be mainly derived from tree-like vegetation (Vogts et al., 2009; Garcin et al., 2014; Zhang et al., 2021), whereas the $C_{31}$ reflect a mixed signal from trees and grasses. The distribution patterns of the contributing plant types are closely mirrored in topsoils (Carr et al., 2014). However, some studies have identified limitations of this approach and question the validity of distinctions by distribution patterns alone (e.g., Bush and McInerney, 2013; Jansen and Wiesenberg, 2017). Therefore, it is recommended to combine the information derived from *n*-alkane distribution patterns with additional proxies, such as compound specific isotopic compositions (e.g., Hockun et al., 2016; Herrmann et al., 2016; Zhang et al., 2021).

## 2.3 Plant-wax compound-specific isotopes

The carbon and hydrogen in plant-wax *n*-alkanes derive from the carbon and hydrogen source the plants utilize for biomass metabolism, i.e., inorganic carbon and water. The isotopic composition of *n*-alkanes thus allows interpretation of either metabolic process during biosynthesis or of effects influencing the isotopic composition of the respective carbon and hydrogen sources (Sachse et al., 2012; Sessions, 2016; Diefendorf and Freimuth, 2017).

In the case of carbon, the effect of different photosynthetic metabolic pathways of different types of vegetation has the greatest influence on the compound-specific stable carbon ($\delta^{13}$C) of $n$-alkanes. Information on contributing photosynthetic plant types can be obtained from their compound-specific $\delta^{13}$C isotope compositions (Diefendorf and Freimuth, 2017). Waxes from $C_4$ vegetation are isotopically enriched in $^{13}$C relative to waxes from $C_3$ plants due to a more effective photosynthetic carbon ($CO_2$) fixation (Collins et al., 2011). Most tropical grasses are of $C_4$ type while all trees and shrubs are $C_3$ plants. The occurrence of $C_4$, i.e. $^{13}$C-enriched, waxes, in sediments has thus been attributed to contributions from grassy environments. However, plants from specific environments, such as specific *Cyperaceae* or *Poaceae* in swamps or salt-tolerant plants in saline settings, can also be of $C_4$ type, complicating a simple interpretation of $\delta^{13}$C compositions (Schefuß et al., 2011). A complication arises when CAM plants are present. CAM plants produce variable and intermediate $\delta^{13}$C compositions but occur only in specific environments, such as in the Succulent Karoo Biome along the west coast of South Africa (Carr et al., 2014). The $\delta^{13}$C compositions of plant waxes also depend on environmental stress, such as water shortage. Under drought conditions, plants close their stomata to increase their water use efficiency leading to an $^{13}$C enrichment (Hou et al., 2007). $\delta^{13}$C compositions of plant waxes should thus only be interpreted in conjunction with other parameters.

The hydrogen isotope composition ($\delta$D) of plant waxes mainly depends on environmental controlling factors rather than biosynthetic mechanisms. The hydrogen that plants incorporate into their biomass originates from the water they absorb. The hydrogen isotope composition of this water mainly depends on the hydrogen isotope composition of precipitation and the hydrological status of the environment. Regarding the hydrogen isotope composition of rainfall different effects can be distinguished (Gat, 1996). The continental and altitude effects describe the isotopic depletion with progressive rainout when moisture travels inland or uphill. The temperature effect describes the isotopic depletion with lower condensation temperatures. For the Mkhuze system, these effects result in plant-waxes being isotopically depleted in hydrogen when derived from more inland or mountainous regions whereas temperature variations play a negligible role. In contrast, the amount effect has a large influence leading to depleted hydrogen isotope compositions of rainfall under high precipitation regimes (Dansgaard, 1964). Wetter areas are thus characterized by more depleted hydrogen isotope compositions of plant waxes. The latter effect is amplified by secondary isotope effects due to evaporation and transpiration. Water in soils, lakes, rivers and wetlands may be isotopically enriched due to evaporation while also leaf water used for biosynthesis of waters may become isotopically enriched due to transpiration (Kahmen et al., 2013). Such isotopic enrichment due to evapo-transpiration has, for instance, been observed in dry environments of South Africa (Herrmann et al., 2017). Additionally, a slight dependency of $\delta$D composition of waxes occurs for different plant types (Sachse et al., 2012). Disentangling all processes affecting the $\delta$D of plant waxes may not always be possible in specific environments. Nevertheless, in combination these primary and secondary effects give an indication of the hydrological status of the contributing vegetation, i.e., about the general humidity of environments, especially when considered in conjunction with other parameters, as is done in this study.

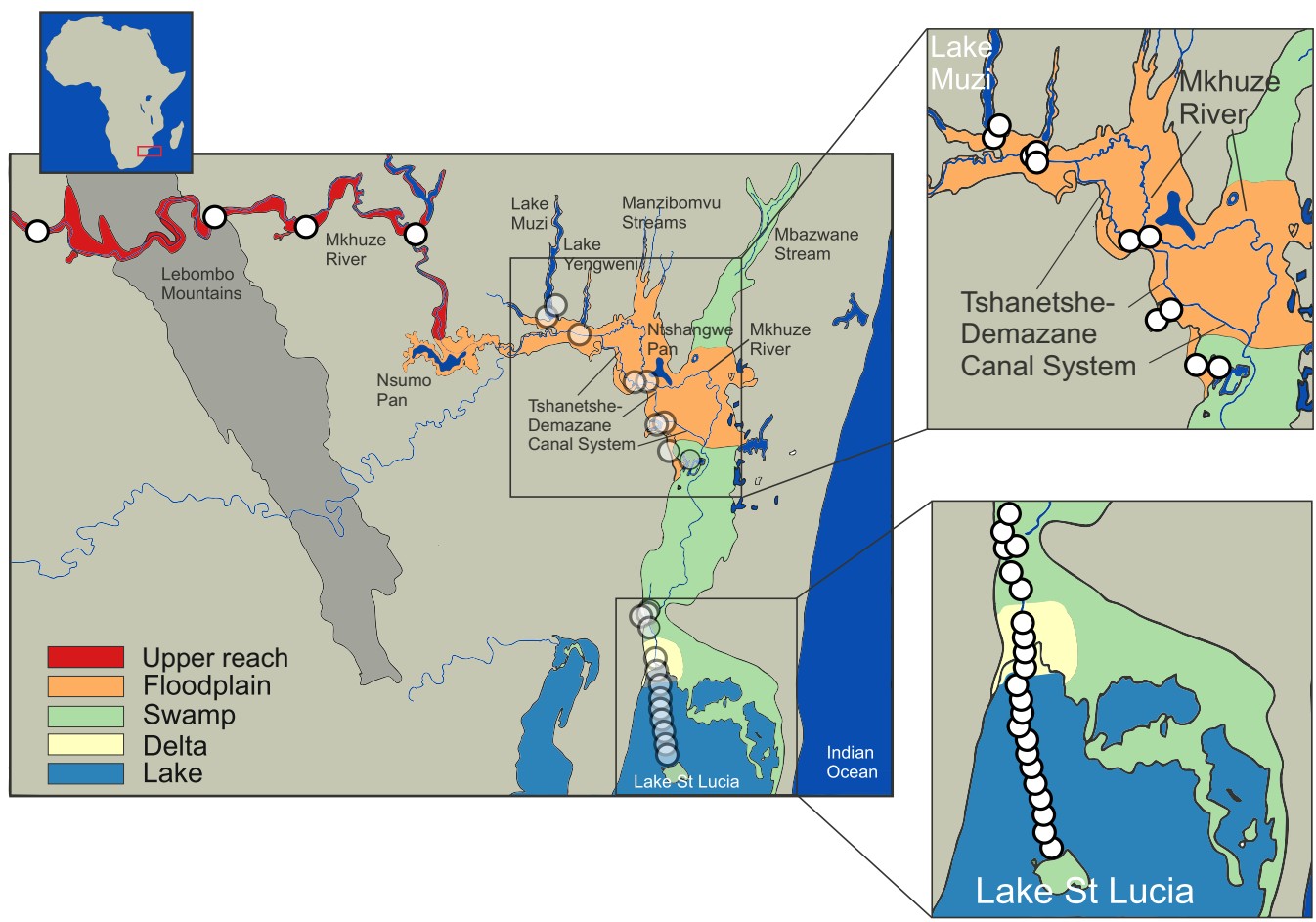

**Figure 1.** Map of the Mkhuze catchment area. White circles represent sampling locations and coloured areas refer to the assigned sub-environments. Important geologic features, including watercourses and major lakes or pans, are named.

## 3    Material and Methods

### 3.1    Study Site

The Mkhuze Wetland System (27.8° S, 32.5° E; Figure 1), South Africa's largest freshwater wetland system (~ 450 km$^2$), is located in the northeastern coastal region of South Africa in the KwaZulu-Natal province. It is bordered by the Lebombo Mountain Range to the west and the Indian Ocean to the east, and forms a mosaic of wetland types, including swamp forests, grassy swamps, and open water (pans) (Stormanns, 1987). This ecological heterogeneity and biotic diversity are of international conservation importance, and therefore listed as a wetland of international importance by the RAMSAR Convention.

### 3.1.1 Hydrology, vegetation and depositional sub-environments.

Two major hydrologic inflows supply water to the Mkhuze Wetland System, the Mkhuze River and Manzimbomvu streams, either through local runoff, direct precipitation, or groundwater inflow (Ellery et al., 2003). The major tributary is the Mkhuze River, which originates east of Vryheid (~ 125 km inland) and drains a catchment area of about 5250 km$^2$ (see Figure 2) consisting mainly of Cretaceous-Quaternary aged sedimentary cover of the coastal plain (McCarthy and Hancox, 2000). The perennial river is fed primarily by direct precipitation and is characterized by a mean annual discharge of about 211 to 326 x 10$^6$ m$^3$ (Hutchison and Pitman, 1973). Water flow varies greatly seasonally and interannually, but is generally highest during the austral summer months and lowest to nonexistent during the winter months (McCarthy and Hancox, 2000). The river drains sedimentary strata of the Dwyka, Ecca and lower Beaufort Groups (Karoo Supergroup), as well as Pongola granites and rhyolites in the Lebombo Mountains (McCarthy and Hancox, 2000). It transports comparatively high amounts of sediment dominated by fine, kaolinitic clays originating from Pongola granites and Karoo Supergroup sedimentary rocks (McCarthy and Hancox, 2000). The second hydrologic input is water from the groundwater-fed streams from the north (Manzimbomvu Streams) (Stormanns, 1987; Barnes et al., 2002). Unlike discharge in the Mkhuze River, these streams are characterized by regular, persistent flow and transport negligible amounts of suspended sediment. The Mkhuze Wetland System can be divided into various sub-environments reflecting different depositional and geomorphological characteristics, namely the upper reach, the floodplain, the Mkhuze Swamps, and the outlet to Lake St. Lucia. The upper reach (Figure 1, reddish colour) is the section extending from east of the Lebombo Mountain Range to the Nsumo Pan. It is dominated by trees, such as *Acacia xanthophloea* and *Ficus sycomorus* along the river course (Taylor, 1982b). The Mkhuze River in this area is a degrading river being largely confined to its channel (Alexander et al., 1986). Farther downstream, the river traverses a sandy coastal plain where it forms an extensive floodplain (Figure 1, orange colour) which is characterized by a variety of vegetation communities (*Phragmites mauritianus* reed swamp community, *Imperata cylindrica* hygrohilious grassland community, *Echinochloa pyramidalis* backswamp community, *Ficus sycomorus* riparian forest community distributed along the Mkhuze River, *Cynodon dactylon* floodplain community, *Acacia xanthophloea* woodland community, and the *Nymphaea sp* aquatic community; Tinley, 1959; Neal, 2001, Figure 3) and partly used for agricultural purposes. During periods of high flow, the Mkhuze River overtops its banks and inundates the floodplain (Alexander, 1973). Flooding results in sediment deposition in the immediate vicinity of the channel due to slower flow velocity caused by riparian vegetation, resulting in the formation of natural levees (Stormanns, 1987; Neal, 2001; Ellery et al., 2011), as well as the recharge of pans located in the north-south oriented fossil dune system. As a result, the Mkhuze River is a highly dynamic and rapidly aggrading system, characterized by relatively high sedimentation rates (0.25 - 0.50 cm/yr; Humphries et al., 2010). Loss of water to the surrounding floodplain results in marked reductions in downstream channel size and the Mkhuze River gradually loses definition before terminating in a large freshwater swamp, termed the Mkhuze Swamps (Figure 1, greenish colour). The Mkhuze Swamps function as an intermediate reservoir that fills with water during the summer and releases it into Lake St. Lucia (Figure 1, bluish colour) from the beginning of the dry season when the lake is no longer primarily fed by direct precipitation (Alexander, 1973; Taylor, 1982b). Dominant vegetation of the Mkhuze Swamps include *Cyperus papyrus*, *Phragmites mauritianus*, and *Echinochloa pyramidalis* (Taylor, 1982b). On the

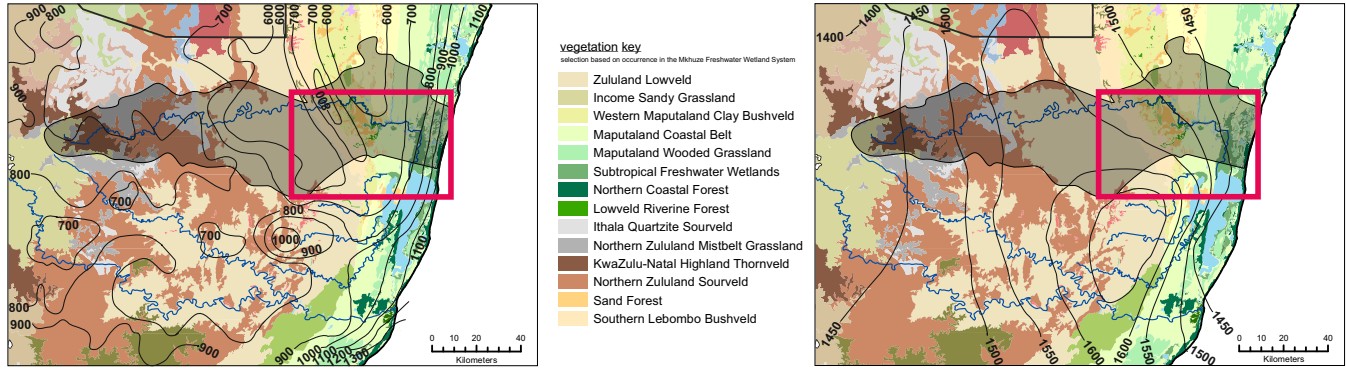

**Figure 2.** The vegetation coverage in the surroundings of the Mkhuze Wetland System is shown. The figure displays the entire catchment area of the Mkhuze River (represented by the outlined, lightened area). The red rectangle indicates the extent of the study area (see Figure 1). In addition to vegetation cover, precipitation isolines (left figure) and evaporation isolines (right figure, in mm/year) are overlaid (adopted from Taylor, 1982a).

levees *Ficus sycamorus* and *Ficus trichopoda* occur, whereas the pans are characterized by *Nymphacaea sp* (Taylor, 1982b). In the northern parts of Lake St. Lucia nearly monospecific stands of the salt-tolerant *Sporobolus virginicus* can be found (Stormanns, 1987). Because Lake St. Lucia is an extremely shallow lake (average depth ~1 m), it is highly susceptible to evaporative losses (~1380 mm/year, Hutchison and Pitman, 1973) and dependent on constant water supply from the swamp system.

### 3.1.2 Climate

The study area experiences a subtropical climate characterized by hot, humid summers and mild, dry winters and mean annual temperatures ranging from 21 °C to 23 °C (Rebelo and Low, 1996). The Mkhuze Wetland System lies within the summer rainfall zone of South Africa, with about 60 % of precipitation occurring during the austral summer months (November through March) in association with cold fronts moving northward along the coast (Watkeys et al., 1993). Precipitation gradually decreases from east to west (see Figure 2, Van Heerden and Swart, 1986) from 1000 mm/year to 600 mm/year (Hutchison, 1976; Maud, 1980). Flooding is highly variable and usually associated with cutoff low pressure systems that develop during December and January, or infrequent tropical cyclones. Evapotranspiration rates are considered relatively high, ranging from 80 mm per month in winter to 190 mm per month in summer (Watkeys et al., 1993).

### 3.1.3 Man-made changes

The course of the Mkhuze River has been greatly altered by human intervention, beginning in the early 1970s. A prolonged drought (1968 - 1971) resulted in hypersaline conditions in Lake St. Lucia (Ellery et al., 2003). The authorities attempted to increase the fresh water supply to the lake by excavating a canal (Mpempe Canal from near Mpempe Pan to 1 km south of Demazane Pan) (Neal, 2001). Flooding caused by Cyclone Domoina in 1984 resulted in severe erosion and the formation of a new stream between Tshanetshe Pan and Mpempe Pan (Taylor, 1986). Additional dredging of a channel (Tshanetshe Canal)

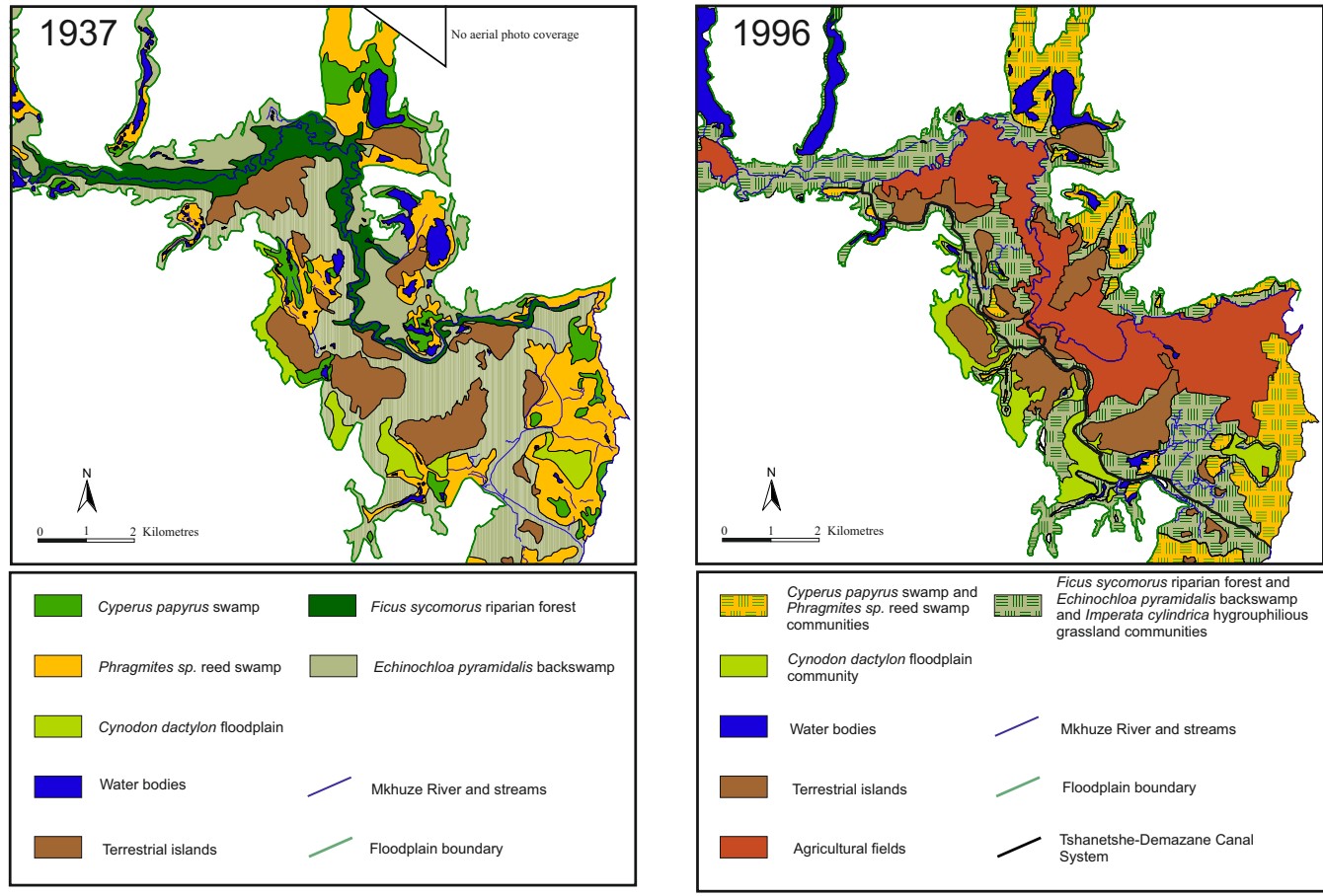

**Figure 3.** Distribution in the vegetation communities and land-use on the Mkhuze Floodplain in 1937 (left) and 1996 (right) (adopted from Neal, 2001).

in 1986 by a local farmer (Neal, 2001) resulted in the fact that today much of the Mkhuze River water is diverted through the Tshanetshe-Demazane Canal System (Stormanns, 1987; Neal, 2001; Barnes et al., 2002; Ellery et al., 2003). Scientific evaluation of the actions taken and their consequences for the system has been overwhelmingly negative (e.g., Alexander, 1973; Taylor, 1982b). However, Ellery et al. (2003) not only provide a detailed description of the channelization processes in the Mkhuze Wetland System to which the reader is referred, but also emphasizes that the alteration of the Mkhuze River flow would also likely have occurred naturally. One aspect which is referred to the channelization processes is a change in vegetation cover. It is reported that formerly extensive stands of *Cyperus papyrus* within the floodplain area were reduced in extension and partly replaced by species which are tolerant to frequent inundation instead of permanently flooded conditions, such as *Cyperus natalensis* and *Echinochloa pyramidalis* (Stormanns, 1987; Neal, 2001).

## 3.2 Sampling

Collection of samples took place during a single field campaign in November/December 2018. Ten plant samples were collected. If possible, replicate plant species were sampled at various sampling sites within the system. The different species were selected based on the occurrence of large cohorts in the field or based on high reported occurrences in previous studies (Stormanns, 1987; Neal, 2001, see also Figure 3), but not all major plant communities mentioned could successfully be sampled during the field campaign. Sampled species include aquatic plants from the *Nymphaceae* family (n = 2) as well as the aquatic plant *Phragmites australis* (n = 2) growing on both dry and flooded soils, two species of wetland grasses, namely *Vossia cupidata* (n = 2) and *Cynodon dactylon* (n = 2), and two representatives of the *Cyperaceaea* family, namely *Cyperus papyrus* (n = 1) and *Cyperus alternifolius* (n = 1). A total of 41 surface sediment samples (uppermost 10 cm) were collected along the course of the Mkhuze River and a transect extending into North Lake (northern part of Lake St. Lucia, Figure 1). The collection of sediment samples was conducted under permit from Ezemvelo KZN Wildlife and iSimangaliso Wetland Park Authority. Because of the limited number of samples due to the difficult accessibility of the sampling area, the results should be considered qualitatively rather than quantitative.

## 3.3 Laboratory Analyses

All surface sediments were freeze-dried. A sieved ($< 212\,\mu$m) sub-sample of approximately 100 mg was removed for classical bulk parameter analysis and the residual sample was ground and sub-sampled for analyses. All glassware was combusted at 450°C for 5h prior to use.

### 3.3.1 Bulk organic matter analyses

The dry, sieved surface sediment was re-wetted with Milli Q water and approximately 20 mL of hydrochloric acid (10 %) was added. After 12 hours, the acidic solution was decanted and the sample was centrifuged at 1000 rpm for 5 minutes. The supernatant liquid was then removed with a pipette. The washing procedure was repeated (4 to 5 times) until a pH of 5 was reached, and the decalcified sediment was freeze-dried. Bulk organic analyses for determining total carbon and nitrogen content and bulk carbon isotope signatures were performed at MARUM, University of Bremen. About 10 mg of decalcified samples were wrapped in a tin capsule and analyzed with a continuous-flow elemental analyzer-isotope ratio mass spectrometer (ThermoFinnigan Flash EA 2000 coupled to a Delta V Plus IRMS). The combustion oven (filled with quartz wool, chromium oxide, and silvered cobaltous-cobaltic oxide), in which C- and N-containing compounds are oxidized, was operated at 999 °C. This was followed by reduction of the resulting nitrogen gases in the reduction reactor (filled with quartz wool and copper reduced granulate) operated at 680 °C. Water formed was removed in a water trap (filled with magnesium perchlorate). Finally, $N_2$ and $CO_2$ were separated chromatographically (using an IRMS steel separation column for NC; length 300 cm, OD 6 mm, ID 5 mm, kept at 40 °C) and transferred on-line to IRMS via a Conflow IV interface. Helium as carrier gas and oxygen as oxidation reagent were used at flow rates of 100 ml/min and 200 ml/min, respectively. Primary standardization of the Delta V Plus IRMS was based on duplicate injections of a reference gas standard from a laboratory tank. The $CO_2$ reference gas

($\delta^{13}C = 34.17\text{‰} \pm 0.1\text{‰}$ vs Vienna Pee Dee Belemnite (VPDB), $5.0\,\text{V} \pm 0.5\,\text{V}$ at m/z 44) was calibrated using IAEA-CH-6 international standards. Quantification of total nitrogen and organic carbon was achieved by external standard calibration using peak areas that yielded a linear 5-point calibration curve. Repeated analysis of an internal laboratory standard cross-referenced to the certified IAEA-CH-6 international standard yielded a precision and accuracy of $0.2\text{‰}$ each. Sample concentration data are reported blank corrected.

Thermal analyses were performed at IFP Energies Nouvelles Lab (Rueil-Malmaison, France) using a Rock-Eval® 6 device (Vinci Technologie). About $80\,\text{mg}$ of freeze-dried ground sample was pyrolyzed in an inert atmosphere ($N_2$) by heating from $200\,°C$ to $650\,°C$ at $25\,°C/\text{min}$, then residual carbon was combusted in air from $300\,°C$ to $850\,°C$ at $20\,°C/\text{min}$ (Espitalie et al., 1985; Disnar et al., 2003). Gases released were monitored by a flame ionisation detector (FID) for hydrocarbon compounds (HC), and by infrared detectors (IR) for CO and $CO_2$. Total Organic Carbon (TOC in wt-%), Mineral Carbon (MinC in wt-%), Hydrogen Index (HI in mg $HC/g\,TOC^{-1}$) and Oxygen Index (OI in mg $CO_2/g\,TOC^{-1}$) were calculated by integrating the amounts of HC, CO, and $CO_2$ produced during thermal cracking and combustion of OM ort thermal decomposition of carbonates between defined temperature limits (Behar et al., 2001; Lafargue et al., 1998). Since cracking temperature of organic compounds depends on their structural stability, the thermal status of OM was characterized by combining R-index (i.e., relative contribution of most thermally stable HC pools) and I-index (i.e., ratio between thermally labile and resistant HC pools; details in Sebag et al., 2016). As derived from a mathematical construct, if the gradual decomposition of labile compounds is its main driver, OM composition can be described as a continuum from biological tissues to a mixture of organic constituents derived from OM decomposition and plotted along a linear regression line (called "Decomposition line"; Malou et al., 2020) in the I-index vs R-index diagram (called thereafter I/R diagram; Albrecht et al., 2015). However, situations with OM mixture from different sources or where decomposition is so intense that it even affects the more thermally stable pools may generate a distribution diverging from the "Decomposition line". In addition, since decomposition temperature of carbonates depends on their composition, examination of $CO_2$ and CO thermograms enables to identify the carbonate minerals present in the mineral matrix (Pillot et al., 2014; Sebag et al., 2018).

### 3.3.2 Determination of *n*-alkane concentrations

Extraction of plant-wax-derived *n*-alkanes from surface sediments was carried out using a DIONEX Accelerated Solvent Extractor 200 (3 cycles: 5 minutes, $100\,°C$, $1000\,\text{psi}$). The internal standard (ISTD) squalane was added prior to extraction. A mixture of dichloromethane and methanol (9:1, v:v) served as extraction solvent. Recovered total lipid extracts (TLEs) were desulphurised with activated copper for $12\,\text{h}$. Plant samples were cut into small pieces resembling all parts of the plant and extracted by using ultrasonic homogenizers and a sequence of methanol, dichlormomethane:methanol (1:1, v:v), and dichloromethane, respectively, as extraction solvents (five minutes each at room temperature). Solvents were combined and the ISTD was added. Saponification of all TLEs was carried out using $0.1\,\text{M}$ potassium hydroxide solution in methanol:water (9:1, v:v) at $85\,°C$ for two hours and neutral fractions were extracted with hexane. The apolar fraction was retrieved by column separation using deactivated silica (1 % $H_2O$ in hexane) and hexane followed by removal of unsaturated components via column chromatography on silver nitrate impregnated silica using hexane to yield the saturated hydrocarbon fraction.

Analyses were performed using a FOCUS gas chromatograph coupled to a flame ionization detector (GC-FID). The GC oven hosted a Restek Rxi-5ms capillary column (30 m x 250 μm x 0.25 μm). The inlet temperature was set to 260 °C and splitless injection mode was used. The GC oven was set at 60 °C, held for two minutes, increased to 150 °C with a heating rate of 20 °C/min, subsequently followed by an increase of 4 °C/min to the final temperature of 320 °C, which was held for 11 minutes. Quantification of the long-chain $n$-alkanes was performed by external standard calibration using the peak areas. The external standard used for this purpose contains $n$-alkanes ($C_{19}$ to $C_{34}$) at a concentration of 10 ng/μL each and was measured repeatedly after all six samples, achieving a relative standard deviation of< 9.2 %. A blank sample containing only the internal standard (ISTD) and a double blank sample not containing the ISTD were also measured to ensure that no contamination occurred during the sample preparation and measurement.

### 3.3.3 Stable isotope analyses of *n*-alkanes

Compound-specific $\delta^{13}C$ values of the long-chain $n$-alkanes were determined using a TRACE GC Ultra equipped with an Agilent DB5/HP-5ms capillary column (30 m x 250 μm x 0.25 μm) coupled to a Finnigan MAT 252 IRMS via a combustion interface (operation at 1000 °C). The GC oven temperature was set at 120 °C, held for three minutes, and raised to the final temperature of 320 °C at a heating rate of 5 °C/min, held for 15 minutes. $CO_2$ was used as the reference gas. All samples were measured in duplicate if sufficient material was available, and values are given in ‰ VPDB. Standard deviations of replicate analyses of all odd-numbered $n$-alkanes analyzed ($C_{23}$ to $C_{35}$) were less than 0.25 ‰ ($C_{23}$: 0.10 ‰ ± 0.07 ‰, $C_{25}$: 0.07 ‰ ± 0.05 ‰, $C_{27}$: 0.08 ‰ ± 0.06 ‰, $C_{29}$: 0.06 ‰ ± 0.06 ‰, $C_{31}$: 0.09 ‰ ± 0.06 ‰, $C_{33}$: 0.07 ‰ ± 0.06 ‰, $C_{35}$: 0.08 ‰ ± 0.06 ‰). Accuracy and precision were determined by analyses of an external $n$-alkane standard calibrated against the A4-Mix isotope standard (A. Schimmelmann, University of India) and measured repetitively every six samples. The precision (% RSD) and accuracy (bias compared to the offline value determined via elemental analysis) of the internal standard ($\delta^{13}C_{squalane}$ = -19.9 ‰ ± 0.3 ‰) were 1.6 % and -0.27‰, respectively. Compound-specific stable hydrogen isotope $\delta D$ values were determined using a TRACE GC Ultra (column and temperature program are the same as for $\delta^{13}C$) coupled to a Finnigan MAT 253 IRMS via a pyrolysis reactor (operating at 1420 °C). $H_2$ was used as the reference gas and all samples were measured as duplicates when sufficient sample volume was available. Reported values are in ‰ Vienna Standard Mean Ocean Water (VSMOW). Standard deviations of replicate analyses of all $n$-alkanes analyzed ($C_{23}$ to $C_{35}$) were less than 3 ‰ ($C_{25}$: 1.0 ‰ ± 0.59 ‰, $C_{27}$: 1.08 ‰ ± 0.67 ‰, $C_{29}$: 0.49 ‰ ± 0.54 ‰, $C_{31}$: 0.46 ‰ ± 0.46 ‰, $C_{33}$: 0.62 ‰ ± 0.46 ‰, $C_{35}$: 0.68 ‰ ± 0.56 ‰). Accuracy and precision were determined by analyses of an external $n$-alkane standard calibrated against the A4-Mix isotope standard (A. Schimmelmann, University of India) measured repeatedly every six samples. The precision ( %RSD) and accuracy (bias compared to the offline value) of the internal standard ($\delta D_{squalane}$ = -180 ‰ ± 3 ‰) were 1.4 % and 0.03 ‰, respectively. The H3 factor was repeatedly measured and gave a value of 4.8 ± 0.1 over the whole measurement series.

## 3.4 Distributional parameters of *n*-alkanes

The carbon preference index (CPI) and the average chain length (ACL) were adapted from Cooper and Bray (1963) and Poynter and Eglinton (1990), respectively. The following calculations were made:

carbon preference index: $CPI = 0.5 * \left[ \frac{\sum_{23}^{33} C_{odd}}{\sum_{22}^{32} C_{even}} + \frac{\sum_{25}^{35} C_{odd}}{\sum_{24}^{34} C_{even}} \right]$

average chain length: $ACL = \frac{\sum_{23}^{35} i * C_{i,odd}}{\sum_{23}^{35} C_{i,odd}}$

relative concentration (contribution) [%]: $C_x = \frac{C_x}{\sum_{23}^{35} C_{even,odd}}$

## 3.5 Statistical Analyses

 All statistical analyses were performed using R software (R version 4.0.3 and RStudio version 1.4.1103). To test whether statistically significant differences occurred between the sub-environments, the Kruskal-Wallis test was performed using the stat_cor_mean() function of the "ggpubr" package (version 0.4.0). When the *p* value indicated that differences ($p<0.05$) between sub-environments were evident, the test was supplemented with a pairwise Wilcoxon rank sum test (base package "stats") to determine which sub-environments had significant differences between them. The Benjamini and Hochberg "BH" method  was used to adjust p-values. Correlation coefficients were calculated by using the stat_cor function with the default Pearson method of the "ggpubr" package (version 0.4.0). Box-and-whisker plots were generated using the "ggplot2" system (version 2.3.3.3) of the "tidyverse" package (version 1.3.0) and "ggpubr" (version 0.4.0) and the geom_boxplot() function implemented here. . Here, the median of the respective data is shown as a solid line and the lower and upper hinges correspond to the first (25th percentile) and third (75th percentile) quartiles. The lower/upper whisker extends from the lower/upper hinge to the  smallest/largest value if it is not smaller/larger than 1.5 times the interquartile range from the hinge. Otherwise, the respective data point is drawn as a single point representing an outlier. For the principal component analysis (PCA) performed, missing values of the data set were first imputed using the impute.PCA() function and applying the regularized iterative PCA algorithm of the "missMDA" package (version 1.18). Subsequently, the actual PCA calculations were performed by using the prcomp() function (base package "stats").

 # 4 Results

## 4.1 Chemical parameters of bulk organic matter

Total organic carbon (TOC), as a measure of organic matter (OM) content, increased in surface sediments from the upper reach to the swamp sub-environment, where samples contained the highest amounts of TOC ranging from 0.9 % to 8.1 % (Table 1 and Figure 4). Higher variance is observed for the upper reach, floodplain, and swamp sub-environment, compared to very  narrow ranges of values in the delta (1.4 % - 1.8 %) and lake sub-environment (1.4 % - 1.7 %). Samples from the floodplain

**Table 1.** Chemical parameters of sedimentary bulk organic matter: total organic carbon content (TOC) in % dry weight (DW), carbon to nitrogen ratio (C/N), summed concentration of medium to very-long chain plant wax-derived $n$-alkanes normalized to dry weight and organic carbon (OC), and bulk organic carbon isotopic composition in per mil in reference to Vienna Pee Dee Belemnite.

| sub-environment | TOC [% DW] | C/N (mass) | $\sum conc_{C_{22}-C_{35}}$ [$\mu$g/g DW] | $\sum conc_{C_{22}-C_{35}}$ [$\mu$g/g OC] | $\delta^{13}C_{org}$ [‰] | CPI | ACL |
|---|---|---|---|---|---|---|---|
| upper reach | $1.8 \pm 0.5$ | $11.8 \pm 1.0$ | $7.0 \pm 0.8$ | $280.2 \pm 165.0$ | $-21.87 \pm 3.84$ | $8.6 \pm 0.7$ | $30.4 \pm 0.3$ |
| floodplain | $3.2 \pm 2.6$ | $14.7 \pm 2.7$ | $2.9 \pm 3.4$ | $124.5 \pm 81.7$ | $-19.65 \pm 1.19$ | $6.2 \pm 1.9$ | $30.7 \pm 0.2$ |
| swamp | $4.6 \pm 2.7$ | $14.1 \pm 2.1$ | $4.7 \pm 2.9$ | $114.3 \pm 32.9$ | $-20.86 \pm 0.86$ | $7.5 \pm 0.7$ | $30.6 \pm 0.2$ |
| delta | $1.5 \pm 0.3$ | $11.4 \pm 1.4$ | $1.6 \pm 0.8$ | $89.0 \pm 23.3$ | $-19.66 \pm 0.45$ | $6.8 \pm 1.0$ | $30.4 \pm 0.4$ |
| lake | $1.6 \pm 0.1$ | $12.9 \pm 1.2$ | $1.8 \pm 0.3$ | $119.5 \pm 20.2$ | $-19.36 \pm 0.73$ | $5.5 \pm 0.2$ | $29.8 \pm 0.1$ |

*Displayed are the median and the median absolute deviation of analyzed parameters. For CPI and ACL, the averages and standard deviations are displayed*

and swamp sub-environments show significantly higher C/N ratios compared to the other sub-environments ($p < 0.05$, Table 1). To estimate the total amount of plant wax input to the sub-environments, the sum of the concentrations of all long-chain odd-numbered $n$-alkanes was considered (Table 1 and Figure 4). When normalized to sample mass, delta and lake samples contained significantly lower amounts of plant wax-derived lipids compared to swamp samples ($p < 0.005$). However, when normalized to organic carbon, no significant differences were detected between any of the sub-environments investigated ($p > 0.05$). Bulk OM $\delta^{13}C$ shows more depleted values in the samples from the upper reach and the swamp sub-environments than in the other sub-environments, but statistical evidence is present only for the distinction between the swamp sub-environment and the delta/lake sub-environment ($p < 0.05$).

## 4.2 Thermal analysis of bulk organic matter

The HI values show a decreasing trend from the upstream (ca. 115 mg HC/g TOC$^{-1}$ for upper reach and 100 mg HC/g TOC$^{-1}$ for floodplain; Figure 5 A) to the downstream sub-environments (ca. 75 mg HC/g TOC$^{-1}$ for swamp and delta; ca. 60 mg HC/g TOC$^{-1}$ for lake; Figure 5 A). The range around the mean value is generally small (ca. 25 mg HC/g TOC$^{-1}$), except for floodplain samples which displayed a high variability, ranging between ca. 50 and 135 mg HC/g TOC$^{-1}$, (Figure 5 A). The R-index values show a comparable pattern with decreasing mean values from upstream ($> 0.60$ for upper reach and floodplain; Figure 5 B) to downstream sub-environments ($< 0.60$ for swamp, delta and lake; Figure 5 B). The highest values were measured in floodplain samples ($> 0.65$, Figure 5 B, while swamp samples displayed the highest variance splitting into two groups (ca. 0.55 and 0.65, Figure 5 B). The I-index values revealed an inverse pattern with low values in upstream sub-environments (ca. 0.1 for upper reach and floodplain, Figure 5 C), high values for the downstream sub-environments (ca. 0.3 for delta and lake, Figure 5 C) and swamp samples divided into two groups (ca. 0.06 and 0.22, Figure 5 C).

In the I/R diagram (Figure 6) the studied samples projected onto the "decomposition line" describing the linear relationship between R and I when the gradual decomposition of the most labile constituents controls the OM transformation (i.e., stabilization). Lake and delta samples contain the most labile OM ($R < 0.57$ and $I > 0.25$, Figure 6) while upper reach and floodplain

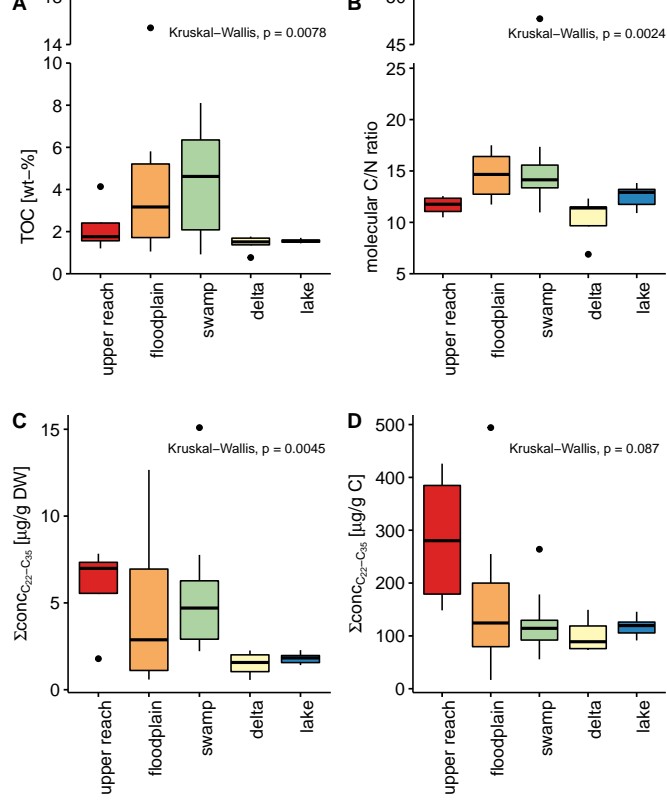

**Figure 4.** Box-and-whisker plots show total organic carbon content as a percent of dry weight (**A**), carbon to nitrogen ratios (**B**), summed concentration of long-chain *n*-alkanes normalized to dry weight (**C**), and summed concentration of long-chain *n*-alkanes normalized to organic carbon (**D**) of all surface sediments. The colours of each box refer to the assigned sub-environments of the Mkhuze Wetland System. *Note that the y-axis of A and B is broken.*

samples contain the most stable OM (R > 0.62 and I < 0.25, Figure 6) with swamp samples falling in between these extremes (Figure 6). It is important to highlight that floodplain and swamp samples display high dispersion around the "decomposition line" in comparison with the strong correlation ($R^2 > 0.9$) usually observed for composts, litters, and topsoils (Albrecht et al., 2015; Sebag et al., 2016).

### 4.3 Distribution patterns and stable carbon isotopic composition of *n*-alkanes

All surface sediment samples analyzed contained long-chain, odd-numbered *n*-alkanes ($C_{23}$ to $C_{35}$). The carbon preference index (CPI) has average values of $6.8 \pm 1.5$ for all surface sediments, confirming that *n*-alkanes originated from plant waxes due to the characteristic odd-over-even dominance. The CPI of the collected plant samples was $11.4 \pm 6.4$. The average chain length

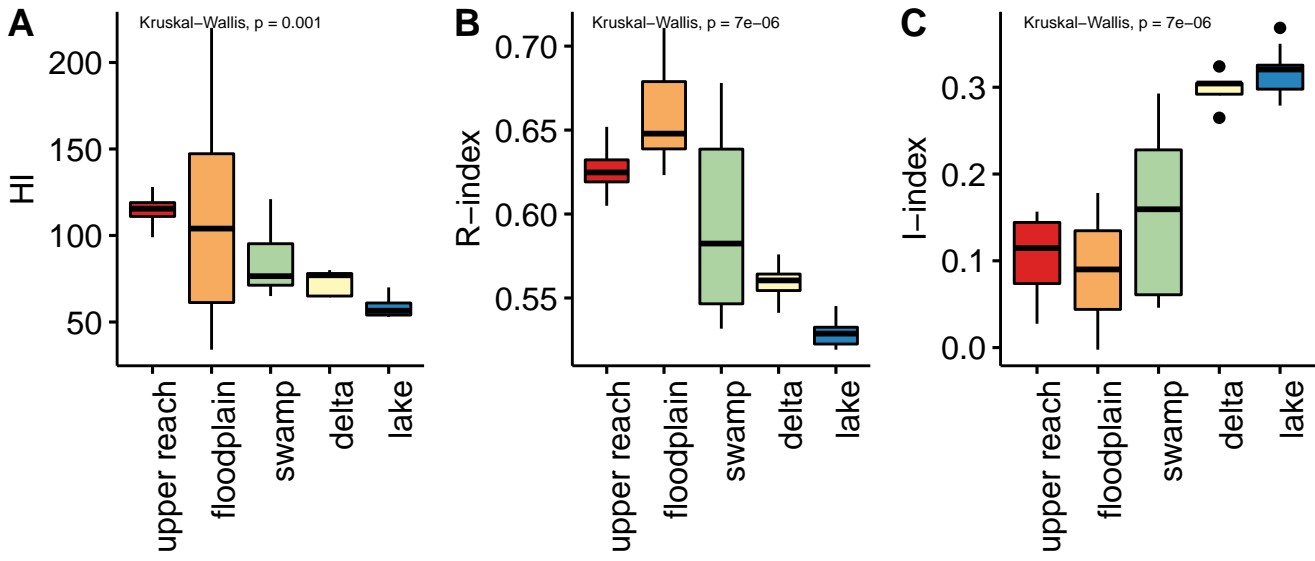

**Figure 5.** Displayed are the indices determined by Rock Eval analyses. Hydrogen Index (HI, **A**), R-index (**B**), and I-index (**C**), as introduced by Sebag et al. (2016) are displayed grouped by assigned sub-environments (see Figure 1 and section 3.1.1) of the Mkhuze Wetland System.

(ACL) for surface sediment samples was $30.2 \pm 0.5$, reflecting the high proportion of longer-chain *n*-alkanes, and $29.6 \pm 1.9$ for plant samples.

### 4.3.1 Plant samples

The sampled plants can be distinguished from each other based on their relative *n*-alkane distribution patterns and corre-
380 sponding $\delta^{13}$C signatures. Aquatic plants belonging to the *Nymphaceaea* family (common water lilies) show a symmetrical distribution around the two dominant *n*-alkanes $C_{27}$ ($31.2\,\% \pm 0.7\,\%$) and $C_{29}$ ($33.0\,\% \pm 4.8\,\%$) (Figure 7 A). Similarly, *P. australis* (common reed), an emergent aquatic plant species, also shows high concentration of the $C_{27}$ *n*-alkane ($22.1\,\% \pm 1.2\,\%$), but the dominance of $C_{29}$ ($41.9\,\% \pm 9.9\,\%$) determines the *n*-alkane distribution pattern (Figure 7 B). Based on their stable carbon isotope signatures, these aquatic plants are classified as $C_3$ plants (*Nymphaceaea*: $\delta^{13}C_{C_{31}}$ = -30.60‰ $\pm$ 0.61‰, *P.*
*australis*: $\delta^{13}C_{C_{31}}$ = -34.99‰ $\pm$ 0.55‰). The $C_4$ ($\delta^{13}C_{C_{31}}$ = -20.53‰) wetland sedge *C. papyrus* (common papyrus, Figure 7 C) shows a symmetrical distribution around the dominant *n*-alkane $C_{31}$ ($31.9\,\%$) accompanied by comparatively similar contributions of the *n*-alkanes $C_{29}$ ($19.5\,\%$) and $C_{33}$ ($12.8\,\%$). It is striking that the sum of the relative concentrations of the long-chain, odd-numbered *n*-alkanes is just over $70\,\%$, the remaining is accounted for by the respective even-numbered *n*-alkanes, of which $C_{32}$ ($15.3\,\%$) has the largest share. The distribution pattern of another *Cyperaceae* species, namely the $C_3$
($\delta^{13}C_{C_{31}}$ = -38.87‰) plant *C. alternifolius* (umbrella papyrus, Figure 7 D), is characterized by predominantly the $C_{31}$ *n*-alkane ($66.1\,\%$). Wetland grasses, such as $C_4$ ($\delta^{13}C_{C_{31}}$ = -18.66‰ $\pm$ 0.21‰) plant *V. cuspidata* (hippo grass, Figure 7 E) exhibits co-

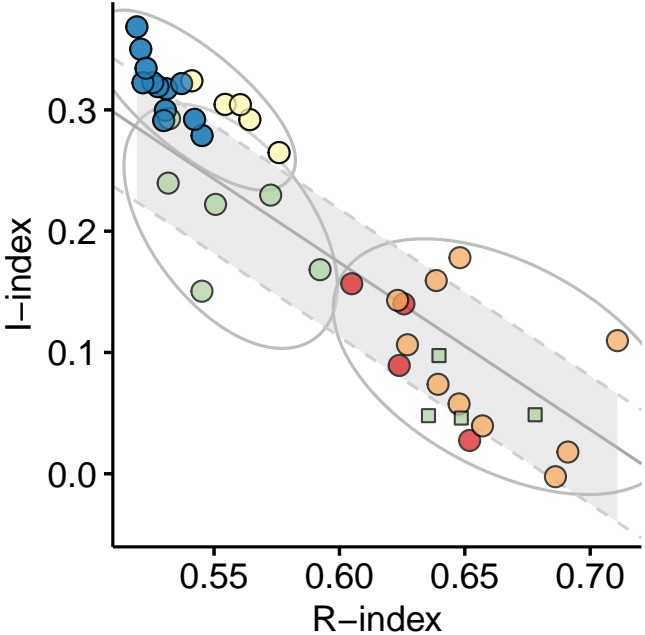

**Figure 6.** I/R diagram. Symbol colours reflect the sub-environment (see Figure 1 and section 3.1.1) origin of the sedimentary organic matter: red = upper reach, orange = floodplain, green = swamp, yellow = delta, and blue = lake. The grey shaded area bordered by grey dashed lines refers to the linear regression describing the continuum from biological tissue to a mixture of decomposition constituents ("decomposition line", Malou et al., 2020). Green circles and squares denote splitting up of swamp samples into groups of differing degradation state, while grey bordered ellipses indicate more general splitting of the data set into groups regardless of their depositional origin.

dominant concentrations of the *n*-alkanes $C_{27}$ (27.9 % ± 0.9 %) and $C_{29}$ (26.8 % ± 2.3 %) homologues going along with the occurrence of the very-long-chain *n*-alkanes $C_{31}$ (11.7 % ± 1.6 %), $C_{33}$ (8.9 % ± 2.5 %), and $C_{35}$ (9.4 % ± 1.5 %). *C. dactylon* (bermuda grass, Figure 7 F) another $C_4$ ($\delta^{13}C_{C_{31}}$ = -22.10 ‰ ± 0.06 ‰) wetland grass is clearly determined by the low dis-
395 persion around the dominant very-long-chain *n*-alkane $C_{33}$ dictating the appearance of the distribution with a 56.7 % ± 8.8 % share ($C_{31}$ = 23.5 % ± 11.8 % and $C_{35}$ = 8.1 % ± 6.5 %).

### 4.3.2 Surface Sediments

Figure 8 shows the relative concentrations of *n*-alkanes and their stable isotopic signatures for carbon and hydrogen in surface sediments within each sub-environment of the Mkhuze wetland. For simplicity, not all individual *n*-alkanes are displayed, but
a reduced representation is shown. For reduction, the individual *n*-alkanes are grouped so that only the parameters of their representatives are shown to illustrate the exemplary trends. This grouping was made on the basis of visual criteria (trends across the wetland system) and verified by statistical means. The results of a principal component analysis (not shown) provide information about variables that contain redundant information. For further validation, the coefficients of the linear correlation

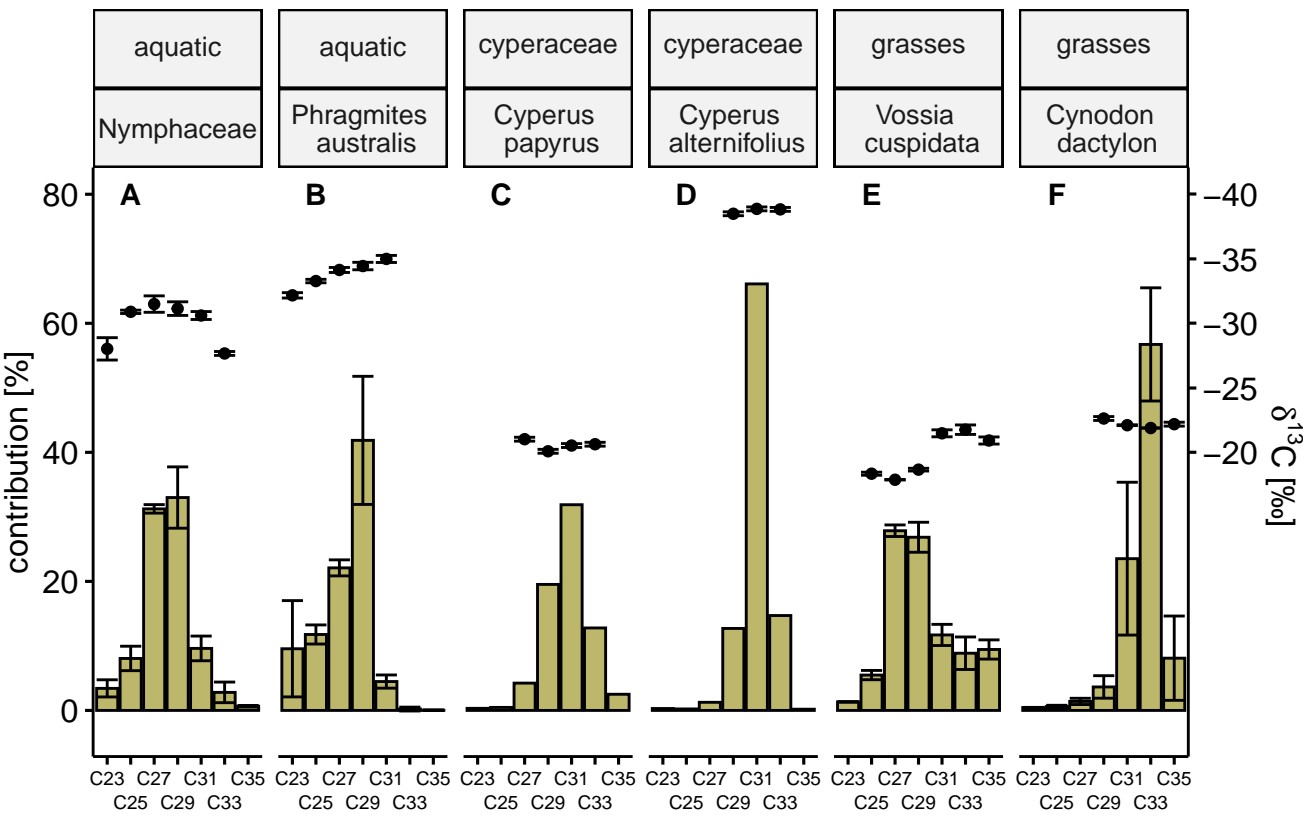

**Figure 7.** Green coloured bars show the relative concentration of long-chain, odd-numbered *n*-alkanes ($C_{23} - C_{25}$) of *Nymphaceaea* **A**, *Phragmites australis* **B**, *Cyperus papyrus* **C**, *Cyperus alternifolius* **D**, *Vossia cupidata* **E**, and *Cynodon dactylon* **F**. Black circles represent corresponding carbon isotope values in reference to Vienna Pee Dee Belemnite (‰ VDPB) of each *n*-alkane. Error bars represent the respective standard deviation. When several representatives of one species were analyzed (A, B, E, F) the error bars reflect natural heterogeneity, while for the others they reflect the analytical uncertainty. When the analytical errors of repeated measurements are below 0.3‰ the intra-laboratory long-term error ($\pm 0.3$‰) is displayed instead. The same routine applies to *n*-alkanes if no repeated measurement was possible due to insufficient sample quantity.

between the individual variables were used. All methods show that the following grouping is justified: (i) the parameters shown for the $C_{25}$ $n$-alkane also reflect trends in the $C_{23}$ $n$-alkane (contribution: $R^2 = 0.72$, $p < 0.001$, $\delta^{13}C$: $R^2 = 0.91$, $p < 0.001$); (ii) the shown parameters of the $C_{29}$ $n$-alkane also reflect the trends of the $C_{31}$ $n$-alkane (contribution: $R^2 = 0.61$, $p < 0.001$; $\delta^{13}C$: $R^2 = 0.86$, $p < 0.001$; $\delta D$: $R^2 = 0.66$, $p < 0.001$); and (iii) the presented parameter trends of the $C_{33}$ $n$-alkane are also representative for the $C_{35}$ $n$-alkane (contribution: $R^2 = 0.55$, $p < 0.001$, $\delta^{13}C$: $R^2 = 0.95$, $p$ textless 0.001). The $C_{27}$ homologue show a correlation with both $C_{25}$ ($\delta^{13}C$: $R^2 = 0.90$, $p < 0.001$; $\delta D$: $R^2 = 0.78$, $p < 0.001$)) and $C_{29}$ ($\delta^{13}C$: $R^2 = 0.90$, $p < 0.001$; $\delta D$: $R^2 = 0.82$, $p < 0.001$) for the stable isotope values, but no stronger correlation in terms of relative contribution to any of the homologues ($R^2 < 0.5$). The medium-chain $n$-alkanes $C_{23}$ and $C_{25}$ show an increasing contribution downstream (upper reach $\rightarrow$ floodplain $\rightarrow$ swamp $\rightarrow$ delta $\rightarrow$ lake), the contribution of $C_{25}$ ranging from $1.5\,\% \pm 0.3\,\%$ in the upper reach sub-environment to $5.4\,\% \pm 0.9\,\%$ in the lake sub-environment (Figure 8 A). The corresponding stable carbon isotope signatures show a $C_3/C_4$ mixed signal (Figure 8 D) with enhanced influence of $C_3$ vegetation in the upper reach, and swamp sub-environments, respectively (Figure 8 D). $\delta D$ values show similar ranges for the upper reach to the swamp sub-environment with a slight decrease in mean values downstream and more enriched values in samples from the delta and lake sub-environment (Figure 8 G). The long-chain $n$-alkanes $C_{29}$ and $C_{31}$ show similar contributions to all sub-environments ($C_{29}$: $13.6\,\% \pm 2.1\,\%$, $C_{31}$: $22.1\,\% \pm 2.1\,\%$), except for a significantly stronger contribution in the upper reach samples ($C_{29}$: $18.4\,\% \pm 2.1\,\%$, $C_{31}$: $28.8\,\% \pm 3.1\,\%$; $p < 0.05$, Figure 8 B). The corresponding stable carbon isotope signatures show a stronger influence of $C_3$ vegetation (Figure 8 E). Decreasing mean $\delta D$ values were observed from the upper reach sub-environment (mean $\delta D = $ -140.8‰, Figure 8 H) to the swamp sub-environment (mean $\delta D = $ -153.3‰, Figure 8 H). The hydrogen isotopic composition of the long-chain $n$-alkanes in the downstream lake sub-environment (Figure 8 H) were in contrast significantly higher (mean $\delta D = $ -133.9‰) compared to the swamp and floodplain sub-environments ($p < 0.05$). The very-long-chain $n$-alkanes $C_{33}$ and $C_{35}$ contribute on average $22.6\,\% \pm 3.8\,\%$ ($C_{33}$) and $6.5\,\% \pm 1.7\,\%$ ($C_{35}$) to all sub-environments, with a maximum contribution in samples from the swamp region ($C_{33}$: $27.6\,\% \pm 7.5\,\%$, $C_{35}$: $8.8\,\% \pm 1.2\,\%$, Figure 8 C). Carbon isotopic values of the very-long-chain $n$-alkanes show a large $C_3$ vegetation influence in the upper reach sub-environment, large variance in the floodplain sub-environment associated with a strong $C_4$ influence that decreases slightly thereafter (Figure 8 F). The hydrogen isotopic composition was similar across all sub-environments, apart from the lake samples which were characterized by elevated values (Figure 8 I).

## 5 Discussion

### 5.1 Plants $n$-alkane distribution patterns as indicators of variable hydrological conditions

Differences in $n$-alkane distribution patterns between plant species have previously been observed in numerous studies (Ficken et al., 2000; Carr et al., 2014; Badewien et al., 2015; Liu et al., 2018). We are aware that the use of $n$-alkane distribution patterns to distinguish plant species and chemotaxonomic fingerprinting approaches are more controversial when transferring findings from one area to another, as it has been shown that variations can also occur within specific species and even between plant parts of the same species (Bush and McInerney, 2013). Because all investigated plants, however, are from the same system and, when possible, multiple plants of the same species were sampled in different sub-environments of the wetland

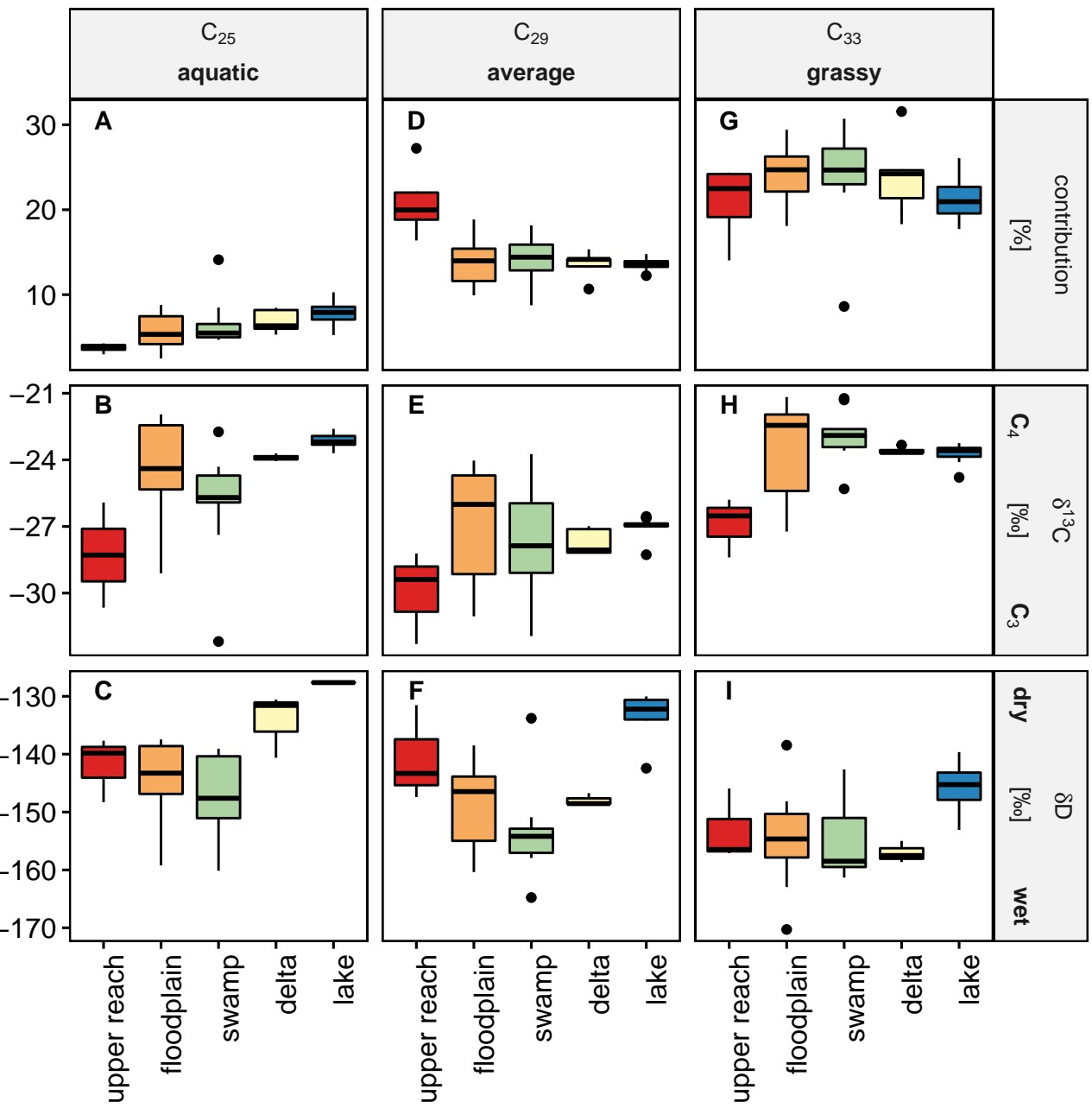

**Figure 8.** Box-and-whisker plots show the relative concentration in % **A**, **D**, **G**, carbon isotope signatures with respect to VPDB in ‰ **B**, **E**, **H**, and hydrogen isotope signatures with respect to VSMOW in ‰ **C**, **F**, **I** of the *n*-alkanes $C_{25}$, $C_{29}$, and $C_{33}$ as representatives for the groups of *n*-alkanes classified as aquatic, average, and grassy.

system, we believe that influences such as variability due to different climatic growing conditions and intra-species variability is small. Although this study investigates a rather small plant sample set, the obtained data shows very good agreement with the literature (see below). For this reason, we are confident that the assignment of specific *n*-alkanes as marker compounds provides realistic indicators of inputs of dominantly occurring plants when performing intra-system comparisons especially when combined with information provided by compound specific isotope analyses.

Aquatic plant species such as the floating *Nymphaceae spp.* (common water lily) and the emergent wetland sedge *P. australis* (common reed) are primarily found in stagnant or slow flowing waters. These two $C_3$ plants (Figure 7 A and B) show elevated concentrations of the medium-chain *n*-alkanes ($C_{23}$ and $C_{25}$), while all other investigated plants show no or negligible concentrations (Figure 7 C to F). The occurrence of the medium-chain *n*-alkanes, although in previous studies reported as predominant alkanes of the respective distribution patterns (Baas et al., 2000; Ficken et al., 2000; Liu et al., 2019), therefore seems to be explicitly indicative of aquatic plant types. The riparian zone along the Mkhuze River is dominated by typical woody plants of riparian forests, such as *A. xanthophloea* (fever tree) and *F. sycomorus* (Sycamore fig) (Neal, 2001). While no representative of these species was sampled in this study, woody plants, such as trees, are generally well studied (Vogts et al., 2009). Tropical $C_3$ trees are typically characterized by high $C_{29}$ and $C_{31}$ *n*-alkane contributions (in sum > 75 %), while the adjacent homologues show concentration mainly below 5 %. Especially the $C_{29}$ *n*-alkane was recently described as "a responsive sensor" of tropical tree vegetation (Zhang et al., 2021) which further corroborates its earlier observed tree-bias (Garcin et al., 2014). However, we pursue to identify alkane homologues which are present in certain plant types and absent or only very low concentrated in others so that they can serve as marker compounds for system-specific vegetation types. For this reason and given the high contributions of this alkane to nearly all displayed distribution patterns, we decided to regard the $C_{29}$ and $C_{31}$ *n*-alkane as more of an integrated signal which possibly originates from a diverse range of plants. The wetland sedges *C. papyrus* (Papyrus; $C_4$) and *C. alternifolius* (Umbrella papyrus; $C_3$) (Figure 7 C and D) predominantly produce $C_{31}$ *n*-alkanes, a finding consistent with the limited data available (Collister et al., 1994). Their natural occurrence is restricted to permanently flooded soils , whereas $C_4$ wetland grasses such as *V. cuspidata* (hippopotamus grass) or *C. dactylon* (Bermuda grass) (Figure 7 E and F) are mostly tolerant of intermittent soil flooding and other disturbances (Holm et al., 1977), so they generally occur widely.Like many (sub)tropical grasses these plant types are characterized by the presence of the very long-chain *n*-alkanes ($C_{33}$ and $C_{35}$) (Rommerskirchen et al., 2006; Vogts et al., 2012), which cannot be found in relevant concentrations in the other investigated plants (Figure 7).

## 5.2 Spatial comparison of organic matter characteristics in surface sediments

Clear differences between the individual sub-environments of the Mkhuze Wetland System become evident by comparing the organic matter characteristics with respect to stability, degree of degradation, primary contributing vegetation and its hydrological growth conditions. That these differences sometimes appear small in absolute values is attributable to the system itself. The discussed sub-environments (see section 3.1.1) cannot be separated from each other by sharp boundaries but rather the transition show gradual transitions in their ecological manifestations. That the characteristics of the OM differ from one another despite this gradual transformation underscores the importance of the individual environments to the system.

### 5.2.1 Upper reach of the Mkhuze River

Bulk organic matter in surface sediments of the upper reach is generally low in concentration (see Table 1 and Figure 4 A) and shows geochemical and thermal characteristics of degraded OM (HI $\cong$ 120 mg/g TOC, C/N = 11.8 $\pm$ 1.0; Figure 5 A and Figure 4 B), which can be related to an advanced decomposition of thermally labile constituents (R $\cong$ 0.63, I = 0.15, linear distribution

within the limits of the "decomposition line" in the I/R diagram, Figure 6). The concentration of *n*-alkanes from plant waxes per dry weight and organic carbon is highest in the surface sediments of the upper reach (Table 1 and Figure 4 C-D) suggesting relative enrichment due to OM degradation. The relative contribution of long-chain *n*-alkanes $C_{29}$ and $C_{31}$ (Figure 8 D) is also highest while carrying the lowest carbon isotopic signature (Figure 8 E) of all sub-environments. Recently, it was shown that especially the $C_{29}$ *n*-alkane might serve as a responsive sensor for tropical tree vegetation (Zhang et al., 2021). Regarding

the upper reaches and despite the $C_{29}$ cluster serving as integrated signal from all plant contribution the strong deviation of relative contribution and carbon isotopic composition of this cluster in comparison with all other sub-environments suggests that sedimentary OM of the upper reach is indeed dominated by inputs of woody plant sources. OM in the surface sediments of the upper reach therefore reflect allochthonous contributions from the hinterland, since the Mkhuze River is primarily lined with riparian forests (Neal, 2001). These trees do not tolerate flooding or water saturated soil conditions so that they are found

on the elevated channel levees. This interpretation of the hinterland as origin is further supported by the stable hydrogen isotopes of the respective *n*-alkanes, particularly evident in $C_{29}$ (Figure 8 F). The relatively enriched $\delta$D values suggest that the plants producing them were exposed to relatively dry conditions during their growth phase. These dry conditions are met to a large extent in the hinterland, considering that the precipitation gradient across the Mkhuze Wetland System is nearly halved from 1000 mm/year in the east near the coast to 600 mm/year at the Lebombo Mountains (Maud, 1980, Figure 2).

### 5.2.2 Floodplain

Bulk organic matter in surface sediments of the floodplain is characterized by high variability in both quantity (TOC, Table 1 and Figure 4 A) and quality (HI, R- and I-indexes, Figure 5 A-C). It splits into three groups, the first one corresponds to samples poor in OM (TOC < 1.5 %) and in hydrocarbon compounds (HI < 80 mg HC/gTOC). A second group corresponds to samples rich in OM (TOC > 5 %) and hydrocarbon compounds (HI > 150 mg HC/gTOC). The third group presents an interme-

diate situation (TOC≈2-3 %, HI ≈ 100 mg HC/g TOC). An explanation could consider that this gradual evolution corresponds to a more or less advanced degradation of the sedimentary OM. However, the I-index does not support this interpretation, since the samples with the lowest HI present the least advanced degree of decomposition of thermally labile compounds (I > 0.1), and conversely. In addition, floodplain samples show a wide dispersion in the I/R diagram (Figure 6) which excludes a gradual decomposition. By analogy with previous work on soils, such nonlinear signatures would rather be associated with OM

mixtures of varying quality and origin, which is consistent with such a heterogeneous depositional environment.

    Mixing of OM is further corroborated by the plant wax concentration per dry weight and organic carbon (Table 1 and Figure 4 C-D) showing great variances and therefore that *n*-alkanes are partly relatively enriched during degradation of OM but also inputs of fresh organic material which is in line with elevated C/N ratios (Figure 4 B). High variability is also evident in

the relative contributions of the *n*-alkanes (Figure 8 A, D, G) and the corresponding isotopic signatures (Figure 8 B, E, H) suggesting a mixture of OM of different origin. The floodplain of the Mkhuze Wetland System is characterized by a mosaic of different smaller wetland types, ranging from the locally distinct river channel to seasonally flooded areas to permanent (open) water bodies. The ecologically complex and variable diversity is reflected in all the parameters. However, it is noteworthy, despite this predominant scatter, that the strongest input of $C_4$ vegetation is clearly observed in floodplain sedimentary organic matter (Figure 8 B, E, H). This cannot be explained solely by the occurrence of $C_4$ grasses such as *C. dactylon* or *Echinochloa pyramidalis*, which are recognized as important floodplain vegetation communities (Neal, 2001), since the "grassy" *n*-alkane input is not significantly higher than in the downstream swamp (Figure 8 G). In addition, Neal (2001) describes the floodplain is increasingly used for growing crops, such as the $C_4$ crops sugarcane or corn (Figure 3), which could explain the particularly strong $C_4$ signal.

### 5.2.3 Swamp

The bulk organic matter in surface sediments of the swamp sub-environment is similarly characterized by high variability in both quantity (TOC, Table 1 and Figure 4 A) and quality (HI, R- and I-indexes, Figure 5 A-C). As for floodplain samples, swamp samples show high dispersion depending on their OM and hydrocarbon content with both contents decreasing downstream suggesting a more or less advanced degradation of the sedimentary OM. Similar to the floodplain sub-environment, the I-index oppose this assumption, as the samples with lowest HI (HI < 75) go along with the least advanced degree of decomposition (I > 0.2) pointing towards a mixture of OM related to overprinting of fluvially introduced OM by in-situ produced OM. The I/R diagram, while samples being globally aligned with the "decomposition line", also reveals two distinct clusters (Figure 6 green circles and squares) showing a degraded and much less degraded signature, respectively. These results indicate that the swamp sub-environment contains a wide range of situations, as those similar to upstream (upper reach, floodplain) and downstream (delta, lake) sub-environments, and therefore reflects a transitional sub-environment. Furthermore, the organic matter is characterized by comparatively high C/N ratios, i.e., corroborating addition of in-situ produced organic matter to the samples. This is also seen in the summed concentration of *n*-alkanes normalized to dry weight (Figure 4 C) being very high, while normalized to organic carbon (Figure 4 D) being rather low what is attributed to fresh organic matter inputs.

A similar pattern like for the bulk organic matter can also be observed in the plant wax data. While the *n*-alkane distribution closely resembles that of the floodplain (Figure 8 A, D, G), the corresponding carbon isotope signatures (Figure 8 B, E, H) show a slightly lower influence of $C_4$ vegetation, but rather a mixed $C_3$/$C_4$ signal with slightly increasing $C_3$ vegetation influence downstream. This can be explained by the increased input of the locally dominant vegetation into the sedimentary organic matter (overprinting), yet to a moderate extent. The Mkhuze Swamps are dominated by the $C_3$ wetland sedge *P. australis* (Figure 7 B) occurring along the eastern margin and large stands of the $C_4$ wetland grass *C. dactylon* (Figure 7 F) at the western side of the river channel. Limited overprinting of fluvially transported OM by local vegetation is further supported by the slightly lighter hydrogen isotopic composition of the respective *n*-alkanes (Figure 8 C, F, I) indicating only slightly wetter growth conditions in the swamp area compared to the floodplain although water availability is much more persistent within the swamp area.

### 5.2.4 Delta and lake

The organic matter of the lake and delta samples shows striking differences when compared with that of the upstream sub-environments of the Mkhuze Wetland System. The bulk organic matter in surface sediments shows a homogeneous signature with lowest contents in OM (TOC < 1.6 %, Table 1 and Figure 4 A) and hydrocarbon compounds (HI < 80 mg HC/g TOC, Figure 5 A) of all sub-environments, and a high degree of preservation of thermally labile fractions (I > 0.3, partly in the upper limit of the "decomposition line"). These results differ drastically from the OM results of the upstream sub-environments, but they do not reflect aquatic autochthonous contributions (as indicated by the low HI). Although the sources of this OM are probably terrestrial, it is not a detrital (allochthonous) OM, reworked from the catchment area, but rather a proximal (para-autochtonous) contribution.

In contrast to bulk OM, the relative contribution of all marker $n$-alkane clusters (aquatic, average, and grassy) (Figure 8 A, D, G) and their corresponding $C_3/C_4$ signatures (Figure 8 B, E, H) are not significantly different from the upstream sub-environments. This implies that the sources of plant waxes regarding vegetation type are similar or possibly even the same as upstream. However, considering that the scatter of the compound specific carbon isotopic composition correlates with the diversity of contributing vegetation types (Magill et al., 2019) the narrow spread of stable and bulk carbon isotope signatures (Figure 8 B, E, H and Table 1) suggests that a more restricted range of plant types contributes to lake surface sediments. The associated hydrogen isotopic signatures (Figure 8 C, F, I) indicate that the contributing plants, although very similar compared to the upstream sub-environments, experienced completely different hydrological conditions during growth. The $n$-alkanes in the lake surface sediments have a higher median $\delta$D compared to that of the upstream swamp by about 10 ‰ ($C_{31}$) to 20.0 ‰ ($C_{29}$). There are several scenarios which could explain these significantly higher $\delta$D values. Considering that the upstream swamp area and the sampled lake transect are located in the same longitudinal range the variation in sedimentary $\delta$D is therefore reflecting neither variation in precipitation amount nor isotopic depletion due to inland movement of water vapor, like in case of the upper reach sub-environment. Given the geomorphological and climatic characteristics of Lake St. Lucia (shallow average depth, large surface area, strong wind regime, and high evaporation rates, salinity), the higher hydrogen isotope signatures of the sedimentary $n$-alkanes in the lake probably resulted from a dominant contribution of vegetation, which use the lake's water as dominant water source. Considering that not only the hydrogen isotopic composition of sedimentary plant-wax-derived $n$-alkanes, but also the previously discussed bulk parameters show striking differences compared to the swamp area more or less excludes the concept of a contribution of a sole unknown "end-member" which alters all mentioned variables except relative $n$-alkane concentration and their carbon isotopic composition although only minimally contributing. All in all, the most reasonable explanation is that the majority of the organic matter in the lake area is not derived from the upstream sub-environments, although this might be unexpected.

### 5.3 Transport of sedimentary organic matter in the Mkhuze Wetland System

Characterization of organic matter in surface soils and sediments in terms of its stability, degree of decomposition and source vegetation, as well as their hydrological conditions in the sub-environments of the Mkhuze Wetland System, allows us to eval-

uate transport pathways. Plant-wax lipids are hydrophobic and associated with the mineral component of sediments (Hedges and Keil, 1995; Keil et al., 1997; Wiesenberg et al., 2010). Thus, the transport and identification of sources and sinks is not exclusive to the organic component of the sediment. In the Mkhuze Wetland System, sedimentary organic matter derived from the Mkhuze River catchment (hinterland) is deposited primarily on the floodplain, but also reaches the Mkhuze Swamps. Under present conditions, Lake St. Lucia does not receive significant quantities of sedimentary material from the hinterland nor material exported from the Mkhuze Swamps. Cores retrieved from Lake St. Lucia (Benallack et al., 2016) and the Mkhuze River bayhead delta (Humphries et al., 2020) indicate that sedimentary infilling commenced 6000 - 7000 years ago when the main oceanic inlets at Lake St. Lucia sealed in response to rising sea levels during the Holocene transgression. This deposited sediment gave rise to the establishment of the current Mkhuze Swamps which currently act as an effective trap preventing the rapid siltation of the lake. An exception might be large disruptive events (severe droughts or strong floods) which occur occasionally and could induce erosion of the swamp or bypassing of its filter capacity. The high concentration of plant waxes in the upper reach of the Mkhuze River is due to low flow conditions during the sampling campaign. During the spring season, the river typically has low or even no flow (McCarthy and Hancox, 2000), resulting in the deposition of suspended sediment in the riverbed. The upper reach samples collected in this study thus likely reflect the "undiluted" hinterland signal (highly degraded, stable, $C_3$, most likely of woody origin). During periods of high flow, transported fluvial OM is deposited along the river channel and on the floodplain by bank overtopping (Neal, 2001; Ellery et al., 2011). The even greater stability of the organic matter in the floodplain illustrates this depositional process (Figure 5 B). Proportions of the organic matter found in the Mkhuze Swamps are characteristically similar to that of the floodplain, so we assume that some suspended material is transported into and deposited in this sub-environment. This is clearly shown in the R- vs. I-index diagram (Figure 6), which shows several clusters of samples: (i) high R-index, low I-index (upper reach, floodplain, swamp subset), (ii) low R-index, medium I-index (swamp subset), and (iii) low R-index, high I-index (lake and delta samples). Surface sedimentary OM from the delta and, in particular, the lake is significantly different from the upstream sub-environments. As discussed in the previous section, Lake St. Lucia not only experiences a significantly lower input of sedimentary organic matter and plant waxes, the deposited material does also not originate from the upstream sub-environments through which the Mkhuze River flows, but most likely from the shoreline around the lake.

Analysed signals from the lake area are thus local in origin and reflect locally occurring eco(hydro)logical conditions. Characteristics of transported hinterland signals predominate only in the upstream areas (upper reach, floodplain, swamp), although in the floodplain and the Mkhuze Swamps a slight overprinting by locally introduced signals can be observed. Based on these findings, we suggest that the Mkhuze Swamps ultimately capture material transported by the Mkhuze River, and consequently the integrated signals encompassing the river catchment, at least under current climatic and environmental conditions.

### 5.4 Local ecological implications

The identification of the Mkhuze Swamps as the ultimate sink for suspended OM from the Mkhuze River confirms previous studies that the Mkhuze Wetland System, specifically the floodplain and swamp, acts as an efficient filter upstream of Lake St. Lucia (Taylor, 1982b; Stormanns, 1987). The current active filtering and trapping function of high sediment loads, including

605 organic sedimentary organic load from the Mkhuze River, may prevent the otherwise rapid siltation of Lake St. Lucia. OM in the surface sediments of Lake St. Lucia originates primarily from lakeshore vegetation, as indicated by the lake transect data shown in comparison with upstream areas of the system. However, some studies (Taylor, 1982b) assume that the Mkhuze Swamps, in their function as freshwater reservoirs ("sponges") (Alexander, 1973), are also responsible for input of OM, serving as a potential energy source in Lake St. Lucia. Our data on particulate organic matter contradict this assumption showing that

sedimentary particulate OM is presently not transported from the hinterland nor exported directly from the swamp. In contrast, OM export from wetlands occurs primarily through the export of dissolved organic matter (DOC, Cole et al., 2007), which accounts for about 90 % of total OM (Reddy and DeLaune, 2008). In saline waters, like Lake St. Lucia, DOM is likely to flocculate (Ardón et al., 2016). Assuming that OM is exported in dissolved form from the Mkhuze Swamps, it should thus be detectable in the lake transect surface samples, but this was not observed in this study. In part, this could be due to the fact

that a significant porportion of DOC may be removed by sorption onto precipitating oxides when sediments contain substantial amounts of aluminum and iron metal oxides (McKnight et al., 1992), as is the case in upstream sub-environments. With the employed methods we cannot confirm neither any particulate OC nor DOC export from the Mkhuze Wetland System into Lake St. Lucia. The present study is the first examining sedimentary OM transport within the Mkhuze Wetland System, revealing that OM is indeed transported even to the Mkhuze Swampsand not being just deposited near the river channel and on the

floodplain. Three processes might play a role for transport of material into the swamps: (i) the ongoing eastward progression of the floodplain (McCarthy and Hancox, 2000), (ii) transport during severe flood events caused by cyclones and cutoff lows, or (iii) that channelization has had an impact on the transport efficiency. The transport path of the Mkhuze River has been dramatically shortened by channelization (see section 3.1.3). As a result, most of the Mkhuze River water now flows through the Tshanetshe-Demazane Canal System (Stormanns, 1987; Neal, 2001; Barnes et al., 2002; Ellery et al., 2003), which may

have altered sediment transport. A shift in the area of deposition of material transported by the Mkhuze River is likely to affect the local vegetation distribution, i.e. causing a shift in the ecological zones by altered substrate conditions. In general, however, the Mkhuze Wetland System overall appears to exhibit high resilience against natural and/or anthropogenic induced changes. The severe drought of 2016, which led to the drying of large parts of Lake St. Lucia, does not seem to have had any lasting impact on the filtering function of the swamps. Likewise, the establishment of the canal system also does not appear to have

caused lasting damage to the filtering function of the Mkhuze Swamps.

## 5.5 Fate of sedimentary organic matter in wetlands

In comparison with humid region (tropical and temperate) wetlands, the Mkhuze Wetland System exhibits distinctive characteristics that reflect the low ratio between precipitation and potential evapotranspiration characterizing the region (Figure 2). High evaporative demand and transmission losses from the river to the surrounding floodplain, result in marked declines in both

channel width and depth downstream (Humphries et al., 2010). Downstream decreases in discharge and stream power result in a gradual decline in ability of the Mkhuze River to transport particulate material, ultimately terminating in the Mkhuze swamps. Although typically unusual, such downstream changes appear to be distinctive features of wetlands found in sub-humid and semi-arid regions of the world (Tooth and McCarthy, 2007), and is likely an important reason why the Mkhuze Wetland System

acts as such an efficient trap for organic material. Most large tropical and temperate river systems are associated with wetlands (Wetzel, 2001; Ward et al., 2017) along their river courses. For example, studies conducted on the Cuvette Congolaise (Runge, 2007), a large wetland system traversed by the Congo River, indicate that fluvially transported particulate OM signals from upstream sources are seasonally overprinted by storage and release of particulates in and from the wetlands (Hemingway et al., 2016, 2017). Depending on variable climatic conditions throughout the year, the Cuvette Congolaise wetlands may thus show a similar trapping function of transported material than the Mkhuze Wetland System under relatively dry conditions while showing increased export from material at higher water flows (Hemingway et al., 2017). Wetlands may thus switch from a trapping function to export of carbon depending on hydrological conditions and seasonal climatic changes. This differential trapping and export functions of wetlands need to be considered when reconstructing climatic changes based on sedimentary archives recovered from terminal lakes and offshore archives. Depending on the activity and efficiency of wetlands, material from more upstream areas will effectively masked by wetlands depending on their hydrologic state and can even be overprinted be wetland export of OM or OM input from downstream areas. Such a process has been suggested for the transport of OM in the Amazon River system, where Andean material is effectively overprinted by lowland sources from rainforests, floodplains and wetlands (Quay et al., 1992; Blair et al., 2004). In such cases, reconstructing environmental changes in the integrated watershed using offshore archives may thus not be possible. Wetland systems with an active trapping function effectively change the transported OM, so that signals detected in offshore archives instead reflect specific sections of the river catchment. Other sediment related proxies may also be affected by wetland trapping, so such geomorphological settings could have a much stronger influence than is often assumed. Combining environmental analyses with specific markers released from wetlands, on the other hand, allows an assessment of the hydrologic changes that lead to inefficient OM trapping and degradation of wetlands (e.g., Schefuß et al., 2016). In addition, carbon sequestration and the ability of wetlands to act as carbon sinks are considered to play an important role in the global carbon budget. There is concern that global warming may alter the hydrological balance of wetlands, releasing significant amounts of carbon to the atmosphere through direct oxidation processes or to adjacent water bodies through erosion of wetland soils, as has been observed for the Cuvette Congolaise and elsewhere (Hemingway et al., 2017). In certain cases, extreme weather events, which are expected to become more frequent as the global climate changes, have also been shown to promote the release of DOC from wetlands (Rudolph et al., 2020). It is likely that particulate material would also be exported by excessive flooding, as is similarly observed by increased flushing of terrestrial carbon into river systems (Bianchi et al., 2013). Thus, the trapping function of wetlands, overridden by overloading, would just as likely contribute to increased carbon dioxide emissions to the atmosphere through turnover of exported OM in adjacent waters.

## 6 Conclusions

We present a spatial assessment of TOC concentrations, OM composition ($\delta^{13}$C, C/N, HI, R-index, I-index), $n$-alkane distributions, and their respective compound-specific stable carbon ($\delta^{13}$C$_{n-alkane}$) and hydrogen ($\delta$D$_{n-alkane}$) isotope compositions along an approx. 130 km-long transect of the Mkhuze River and plant wax data from locally dominant plant species to constrain the origin and transport pathways of OM through and within the sub-environments of the Mkhuze Wetland System,

South Africa. Our results indicate that degraded OM originating from the hinterland is deposited primarily on the floodplain of the Mkhuze River and partially in the downstream swamp. The Mkhuze Swamps currently efficiently trap OM under low flow conditions, so neither release hinterland material nor export swamp-derived OM to Lake St. Lucia (Figure 9).

The surface sediments in the upper reach show allochthonous inputs from the Mkhuze River basin. OM concentrations are low and show a degraded signature associated with a $C_3$ woody source vegetation, which experienced relatively dry growing conditions. Sedimentary OM in the floodplain and swamp exhibit high variability in both source signatures and degradation status, thus reflecting environmental diversity. A mixture of degraded OM from the hinterland and fresh OM of local origin characterizes samples from the floodplain area. In addition, the most pronounced $C_4$ signatures are encountered, attributed

to agricultural use of the floodplain. OM from surface samples in the Mkhuze Swamps also shows a degraded signature but reflects increasing inputs of local wetland sedges and wetland grasses. In contrast, OM from Lake St. Lucia shows completely different characteristics, such as much lower concentrations and much less degradation due to proximal terrestrial inputs rather than reworked catchment derived or aquatic contributions. Plant wax data confirms these findings, pointing to lake shoreline vegetation as the main source.

This study shows that traversed or terminal wetlands under certain conditions, such as low flow, in this case a result of climatic factors, i.e., evaporation exceeds precipitation, can capture OM so efficiently that transport from upstream areas does not occur and downstream OM originates almost exclusively from the immediate vicinity. We emphasize that such wetlands, as geomorphological features within river systems, can impact environmental studies, which assume watershed-integrated signals based on terminal sediments. In addition, disturbances, e.g., by extreme weather events, which are assumed to become

more frequent under global climate change, are likely to affect the trapping function of wetlands and thus increase export of previously stored OM, leading to an increase in carbon emissions through turnover of exported OM in adjacent waterbodies.

*Code and data availability.*   The research data is available at Pangaea (https://doi.org/10.1594/PANGAEA.935586).

*Author contributions.*   MZ, ES, AH and MH conceptualized the project; MZ and ES directed the project and acquired financial support for

the project leading to this publication; DS characterized the samples with Rock-Eval analysis and interpreted related results; JG performed sample preparation, measurements, processed the data and drafted the manuscript; MH and JG designed figures; JG took the lead in writing the manuscript. All authors provided critical feedback and helped shape it.

*Competing interests.*   The authors declare no competing interests.

*Financial support.*   This work was financially supported by the German Federal Ministry of Education and Research (BMBF, Bonn, Germany) under the project "Tracing Human and Climate Impacts in South Africa" (TRACES) project number: 03F0798A.

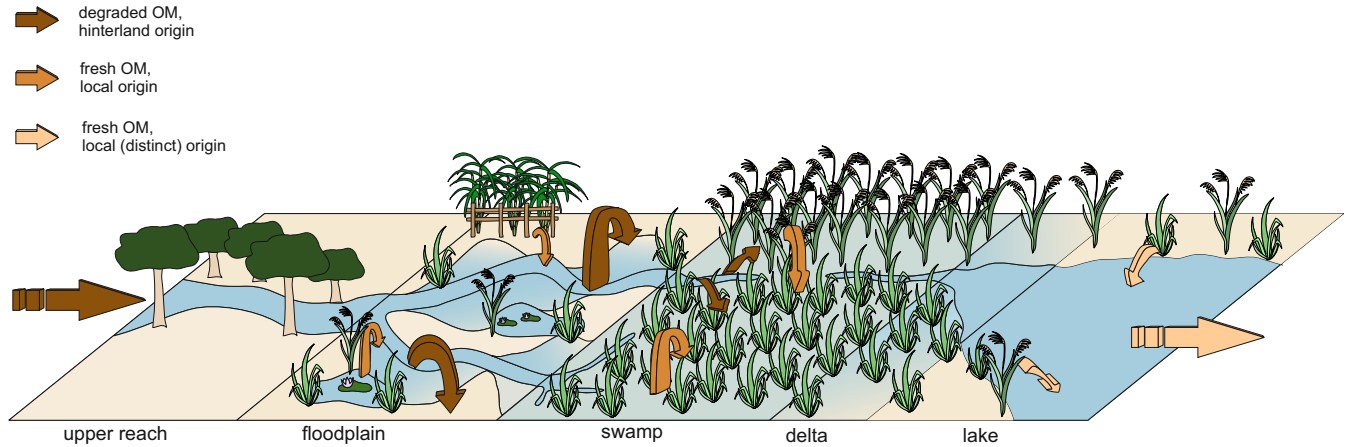

**Figure 9.** The figure shows a schematic summary of the characteristics and transport pathways of organic material in a terminal or flow-through wetland system under low flow conditions. The vegetation types shown (trees, wetland sedges, grasses, and aquatic species) indicate the prominently occurring vegetation in each sub-environment of the Mkhuze Wetland System. Arrows indicate OM input and deposition, with thickness of arrows corresponding to OM quantity assumption and colours corresponding to OM quality characteristics (dark brown: more degraded, light brown: less degraded, spatially different origin).

*Acknowledgements.* We sincerely like to thank Dr. Letitia Pillay, Dr. Archibold Buah-Kwofie, and Lucky for their expertise, support, and energetic assistance in obtaining the material studied. Additionally, we particularly thank Lucky for his assistance in communicating with the local landowners. We are very grateful for the comments by the anonymous reviewers which helped to improve the quality of our manuscript. We thank Dr. Marcus Elvert and Hendrik Reuter for their contributions to previous versions of the manuscript. This study would also not have been possible without the help of technicians Ralph Kreutz and Jenny Wendt of MARUM - Center for Marine Environmental Sciences.
We thank them for their support in chemical analyses and the student assistant Abdullah Saeed Khan to assist in bulk organic matter analyses. We thank Prof. Marion Bamford and Dr. Frank Neumann for their support and help with identifying the plant samples. We are grateful to Daniel Pillot and Herman Ravelojaona (IFPEN) for their technical and scientific support in Rock-Eval® analysis, a trademark registered by IFP Energies Nouvelles. Moreover, we thank the GeoB Core Repository at MARUM and Pangaea (https://www.pangaea.de/) for archiving the sediments and data used in this work. We thank Ezemvelo KZN Wildlife and iSimangaliso Wetland Park Authority for permitting us to
work at Lake St. Lucia. Samples were collected under the registered project: "A multi-proxy investigation into past and present environmental change at Lake St. Lucia".

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
