# Peer review of "Origin, transport, and retention of fluvial sedimentary organic matter in South Africa's largest freshwater wetland, Mkhuze Wetland System"

_Biogeosciences, 2021_

## Referee Comment (RC2)

**Review "Origin, transport, and retention of fluvial sedimentary organic matter in South Africa's largest freshwater wetland, Mkhuze Wetland System" by Gensel et al.**

**General**

The manuscript is generally very well written and focusses on an interesting and important topic, the characterisation of the organic matter input from Mkhuze Wetland System in South African to Lake St. Lucia. The first problem that struck me when reading the introduction was, that aims and questions are very vague., It is not clear what is meant by assessing the status of the wetland (see below). Is the hydroglogical status meant (drained, undrained) or the soil degradation status or the degradation status of OM? Also using $\delta^{13}C$ and $\delta^{2}H$ of n-alkanes to characterize sinks and sources is no doubt a fore front method, but for sure not matured enough, to draw conclusions on plant type communities and degradation status of wetlands.

As such, compared to such ambitious aims, the introduction is very general. I totally miss discussion of state of the art on $\delta^{13}C$ and $\delta^{2}H$ organic matter tracing and what it can tell us about sinks and sources. The same holds true for stability of n-alkane concentrations when used as indicators for sediment or organic source attribution.

The description of the sampling concept is totally missing. All it states is, that "ten samples where collected." However, Figure 1 displays around 30 sampling sites, so I assume that ten samples for each sub-environment was taken? This is totally unclear. A detailed map of vegetation communities is presented (Figure 3) but it is not at all clear, if all these communities were sampled as possible sources and if so, how many samples, which plant species etc....If the aim is, to track OM in the lake back to these communities, the detailed sampling scheme has to be described.

The results are mainly a listing of all measurements done with differences in numbers and sizes. There is no real information gain for the reader, as none of these results are set into perspective and the discussion does not give a clear link back to these data descriptions. Not even the indices and parameters used are in any way explained in the results section (and only very briefly in the discussion). Variability and differences are hard to assess, as sampling numbers and possible errors are not described. It is not clear if error bars indicated in-field heterogeneity or analytical uncertainty. Figure 8 states that error bars might be intra-laboratory long-term errors.

The discussion is more a descriptive qualitative narrative of differences found in parameters within and between different sub-ecosystems. Indices for evaluation are not adequately introduced and partly interpreted in a wrong way (e.g., that CN ration of OM would be a general indicator of chemical stability). As such, I can not follow conclusions drawn and can not judge if these conclusions adequately assess the results. One example would be the conclusion "Sedimentary OM in the floodplain and swamp exhibit high variability in their source signatures and degradation status reflecting environmental diversity, with samples from the floodplain characterized by a mixture of degraded OM from the hinterland and fresh OM."(line 575-577). With clearly high ongoing and very variable degradation of OM in these systems, concentrations of organic substances can not be used as a conservative tracer. Regarding the isotope tracers used, no un-mixing was done and the values were interpreted in a qualitative way, which is, from my perspective not leading to meaningful conclusions.

All in all, I would judge this work as containing highly valuable and interesting data and results. But description of sampling concept is inadequate and interpretation of data is qualitive with numerous assumptions I am not sure can be hold.

**Introduction**

Generally, well written and interesting to read about the Mkhuze Wetland System. However the aim of "assessing the current status" is very vague to me (line 36). Which status do you mean? Hydrological? Soil degradation? Nutrient status?

I totally miss discussion of state of the art on $\delta^{13}C$ and $\delta^{2}H$ organic matter tracing and what it can tell us about sinks and sources. Also, what about the stability of n-alkane concentrations in these systems? Are you sure you can use these as conservative tracers?

40-42 this assumes that you have species specific tracers

42-45 how can you assess the vegetation type (do you mean plant community?) with $\delta^{13}C$ of n-alkanes?

45 hydrological conditions of what? Of the regime under which the plants grew? Of the soil? E.g., wetland, upland, drained? Not sure you can achieve this with $\delta^{2}H$?

51 this is a big aim, to the assess the status of the wetland systems in terms of its filter function and influence on Lake St. Lucia!

**Methods**

133 what do you mean by "Ten samples were collected." Of what? Ten repetitions within a site?

137 – 139 these plants were not collected? But all others were? Or these are the ones which you did collect?

Figure 4 is basically describing standard analysis and could be moved to supporting information.

**Results**

252 what is HI value?

256 what is R-index?

260 what is I index?

For all errors it is not clear from how many reps they are produced, if repetition at all or if this is analytical error.

**Diskussion and Conclusions**

Paragraphs 343 -349 versus 332 – 338: I am not sure I understand you correctly, but this makes not much sense to me. First, you describe the differences in n-alkane patterns within plants, within sites and between different ecosystems, but then you assume that you can take literature values from generally well studied plants, such as trees as source values to be characteristic for your sites?

The discussion in 4.1. mainly compares the n-alkane concentrations determined in this study in comparison to literature values. But what is the message behind this paragraph?

Paragraph 4.2.: what is is this telling me regarding the aim of your study, the status of your system, what is the aim of this paragraph? What is the connection to your results?

369 the C/N ratio is an indicator of chemical stability? I do not think so?

Paragraphs 4.2.1. to 4.2.4. describe the variation of the different measured parameters in each of the sub-ecosystem types and tries to induce state of degradation of OM or plant origin. This discussion is qualitative and not really set in perspective to literature values.

Section 4.3. (line 467 – 480)  starts with a general description of Mkhuze Wetland System which might be transferred to the methods or the introduction. Or are these statements conclusion from your data? If so, please make the link to your results.

494 – 499 is this general knowledge of literature or is this a conclusion from your results? Please make the link to your data

Some conclusions might be considered speculative.... e.g., from the result that "....the higher hydrogen isotope signature of the sedimentary n-alkanes in the lake probably resulted from a dominant contribution of lakeshore vegetation" (line 464 – 465) the general conclusion is drawn, that "OM in the surface sediments of Lake St. Lucia originates primarily from lakeshore vegetation" (line 501). There is no unmixing of possible source signatures, no quantitative evaluation. This is just one example, which leaves the impression, that conclusions drawn are based on rather qualitative assumptions and might even be speculative.

---

## Author Response (AR1)

Biogeosciences Discuss., referee comment RC1 https://doi.org/10.5194/bg-2021-172-RC1, 2021 © Author(s) 2021. This work is distributed under the Creative Commons Attribution 4.0 License.

[Figure]

**Comment on bg-2021-172**

Anonymous Referee #1

Referee comment on "Origin, transport, and retention of fluvial sedimentary organic matter in South Africa's largest freshwater wetland, Mkhuze Wetland System" by Julia Gensel et al., Biogeosciences Discuss., https://doi.org/10.5194/bg-2021-172-RC1, 2021

Julia Gensel and colleagues have presented a paper that characterizes organic matter and its recent spatial distribution in the Mkhuze Wetland System and its catchment. Overall, the paper is sound, but there are some major issues to clarify, especially concerning the *n*-alkane data.

Dear Anonymous Reviewer #1,

We thank you very much for your very valuable comments. We considered them as very helpful to improve the quality of our manuscript. We appreciate the time you took and especially the attention you paid even on the small details. Below, we will give a point-by-point answer to your comments (all line references refer to line numbers of the marked-up manuscript). An explicit list of all made text changes is provided in the end of the marked-up manuscript version.

Sincerely,
Julia Gensel (on behalf of all co-authors)

1) Overall, I suggest major minor revisions to this paper. Two broad comments I have: The separation in three end-member, i.e. aquatic ($C_{25}$), woody ($C_{29}$) and grassy ($C_{33}$) is not visible in the *n*-alkane pattern of modern plants presented in this study. Please clarify why these end-member were chosen, because the vegetation in the studied area likely shows a different pattern for aquatic and grassy vegetation while woody plants were not investigated.

We clarified the reasons why we decided to use specific *n*-alkanes as exclusive marker compounds for certain vegetation types. To do so, we revised subsection 5.1 ("5.1 Plants *n*-alkane distribution patterns as indicators of variable hydrological conditions") of the discussion part to explain which *n*-alkanes we chose for which group of vegetation (l470f, l475ff, l496f). We also eliminated the word "end-member" from the respective text. In addition, we renamed the former "woody" cluster (now: "average") and discuss the meaning of the $C_{29}$ *n*-alkane in more detail (l484ff).

2) Additionally, it is demonstrated that organic matter in lake surface sediments is dominated by local vegetation. I suggest a more clear statement there: What does this imply for paleoenvironmental and –hydrological studies at the end of the respective section and in the conclusions, where it is stated that this study is of great importance for future studies.

We gave a more precise statement on the implication for future (paleo)environmental studies, as suggested. We added information on this to the end of the abstract (l23f) and rephrased the respective sentence in the conclusions (l748ff).

Specific comments:

3) L. 3ff.: Please rephrase this sentence.

We did and in addition, we revised the abstract (l8ff).

4) L. 7f.: Please indicate which signal is present in the upstream area.

We did by giving information on which signal is present in downstream areas compared to upstream areas (l12f).

5) L. 9f.: […] higher dD values. Compared to what?

We added the requested information (l13).

6) L. 10ff.: I cannot follow these two sentences. First you state that lake surface sediments are dominated by local vegetation incorporating a local hydrological signal. Afterwards, you state that those sediments integrate hydrological conditions of the whole watershed. Please clarify these contradicting statements.

We clarified our statement (l17f, l20f). We hope that we now clearly state, that due to fluvially transported OM deposition the assumption of watershed-integrated signals is true for most sub-areas of the investigated wetland except for the downstream lake area which show locally influenced signals instead.

7) L. 27: The Mkhuze Wetland System, […], is […].

We edited the sentence as suggested.

8) L. 53ff.: Move this paragraph to line 23-28 to discuss risks and benefits of wetlands in one paragraph.

We did as suggested and were very grateful for this suggestion.

9) L. 60: I suggest a short section introducing to *n*-alkanes and their compound-specific (CS) $d^{13}C$ and dD isotopic composition including their interpretation. Here you could introduce why you choose $C_{25}$, $C_{29}$, $C_{33}$ and you can refer to recent studies analyzing CS $d^{13}C$ and CS dD in topsoils (e.g. Carr et al., 2014, org. geochem., Herrmann et al., 2016, QSR, 2017, org. geochem., Strobel et al. 2020, STOTEN).

We added a new section ("2 Approach and methodological background") (l81ff) to give a more precise introduction to the used methods. We also incorporated references to recent studies analyzing compound-specific isotopes in topsoils (e.g., l136, l138, l141, l173). The explanation why we used which alkanes as marker compounds for specific vegetation types we limited to the revised subsection 5.1 (l461ff) as the choice of marker compounds is related to the investigated plant samples and we therefore consider them as part of the study's results and consequently their interpretation.

10) L. 69: Please avoid one-sentence paragraphs in the whole manuscript.

We did.

11) L. 71ff.: It seems like a word is missing?

We were thankful for your awareness and added the missing word (l189).

12) L. 73: Can you please characterize the river a bit more in detail, e.g. as episodic/periodic/… river system?

The river is perennial and we added the information to line 190. However, it is characterized by highly variable discharge what we state in line 191. As the manuscript addresses the different sub-areas of the wetland system the river crosses, we decided to keep the detailed information on the river's behavior in the respective sub-environments.

13) L. 110: Please provide a reference for this numbers and also for % precip. in the next line.

We did as suggested (l225, l227).

14) L. 119ff.: Please provide references for all these information.

We did as suggested (l234, l236).

15) L. 131: Can you please provide information about the potential natural vegetation in your studied area?

To the best of our knowledge, the vegetation cover shown in Figure 3 (right subfigure) is the oldest available information on the vegetation of the system, which is also consistent with typical wetland vegetation. Therefore, we are unable to provide any additional information.

16) L. 229ff.: Repetition of "Here" at the beginning of the sentence, please modify.

We did as suggested and removed the complete repeated sentence.

17) L. 270ff.: Is there a table to which you can refer, that the reader can follow this numbers? If not, please provide or add these data to table 1.

We did as suggested and added the data to table 1.

18) L. 290ff.: Can you please refer to a figure? Figure 8?

For both wetland grasses, the reference to the respective sub-figure of Figure 8 is already given after the common name in parentheses (l422 and l425).

19) L. 321ff.: Please check if a minus is missing prior to the dD values.

It was missing and we added it. We also added the other missing minuses (l451, l452, l453). Once again, we are very grateful for the reviewer's attention.

20) L. 339ff.: In figure 8A, B both show a distinct ($C_{27}$) $C_{29}$ dominance. Please clarify.

We revised the subsection "5.1 Plant $n$-alkane distribution patterns as indicators of variable hydrological conditions" to explain more precisely why we chose specific $n$-alkanes as marker compounds for certain vegetation types. Consequently, we also reworded the sentence about the aquatic representatives (l473ff).

21) L. 345ff.: Okay, and how can you distinguish the two woody and grassy end-member when modern grassy plant samples in your study show distinct contribution of $C_{29}$ (figure 8C, E, F)?

We revised the subsection "5.1 Plant $n$-alkane distribution patterns as indicators of variable hydrological conditions" to explain more precisely why we chose specific $n$-alkanes as marker compounds for certain vegetation types. In doing so, we also give more precise information on our approach regarding the grassy cluster (l496f). In addition, we added a more detailed discussion about the $C_{29}$ $n$-alkane (l484) giving information on the evidence that the $C_{29}$ is tree-vegetation biased. But due to our approach aiming for identification of exclusive marker compounds we decided to regard it as a more integrated signal of all vegetation types. Therefore, we renamed the "woody" cluster (Figure 8) and restricted the term "woody" to the respective part of the discussion about samples from the upper reaches (l530ff).

22) L. 353ff.: When there are differences in photosynthetic pathway, what does this mean for the dD signal? Does the photosynthetic pathway has an influence on this signal – see e.g. Sachse et al., 2012. If the abundance of C4 vegetation in the studied area changed during the past, what does this mean for the interpretation of dD in sedimentary record?

Yes, the photosynthetic pathway most likely has an impact on the dD of the n-alkanes, as summarized in the aforementioned review by Sachse at al. We added a sentence regarding this issue with reference to the mentioned review in line 174 in the new subsection 2.3. With respect to our study, the δD comes into play critically when discussing its highly enriched values in the lake area (Figure 8 and subsection 5.2.4). The differences in δD in the lake area are not due to a higher contribution from $C_4$ plants, such as grasses. The data show that both the relative contribution of marker $n$-alkanes and their respective $\delta^{13}C$ isotopic

compositions are comparable with the upstream sub-areas, suggesting a similar or even the same vegetation community (610ff). Since the contributing vegetation does not differ, the difference in δD values must be due to hydrological reasons (differences in source water, evapo(transpi)ration, etc.).

For sedimentary records, vegetation change must be taken into account when interpreting the δD of plant wax-derived *n*-alkanes. Since we have not presented an archive, this answer remains speculative, but we agree with the statement of Sachse et al. that changes in vegetation cover most likely result in either a reduction or exaggeration of differences in sediment δD. If it is possible to track changes in vegetation cover through other proxies (e.g., relative alkane contribution, *n*-alkane- $\delta^{13}$C), we believe it should still be possible to attribute changes in δD to environmental/climate controls.

23) L. 358ff.: This section is more or less a data description, and a discussion of the data is very limited. I suggest to provide a statement if your findings match the expected environmental conditions in each sub-environment.

These paragraphs were intended to provide an introduction and explanation of the approach for the following subsections as they apply to all four. However, we incorporated the information into the new section "2 Approach and methodological background", so that the introduction to the discussion of the organic matter characteristics in the respective sub-areas ("5.2" l500ff) has been streamlined.

24) L. 378: Would the odd-over-even predominance (OEP) of the *n*-alkanes may also provide useful information concerning the state of degradation?

The OEP confirms that the analyzed n-alkanes can be used as conservative tracers since no contamination or extremely advanced degradation processes can be observed (l132, l404).

25) L. 379ff.: Closing bracket is missing. Does this approach really work out at your study site? I agree that this approach is widely applied and I also noticed your statement in lines 332-339, but does this really work in your setting? The data are great, but could you may make this vegetation statement a little more cautious?

The whole paragraph was incorporated into the new section "2 Approach and methodological background" and the statement therefore doesn't exist anymore.

26) L. 387: Please see comments above (C$_{29}$ vs C$_{33}$). Additionally, I suggest to provide a figure showing the *n*-alkane patterns in each sub-environment.

We refined the discussion part about the origin of the C$_{29}$ in the upper reach area (l530ff) and restricted the term "woody" to this particular part of the discussion.
Unfortunately, because the subareas gradually merge into each other rather than being sharply separated (e.g., the contributing vegetation in floodplain, swamp, and lake areas is quite similar), a visual representation of the n-alkane patterns of the subareas is not helpful. The overall similarity would mask the much finer differences, making it difficult for the reader to see and understand our results and interpretations. It would simply weaken our argument, as the human eye would instinctively assume that they all look almost the same, rather than recognizing the differences that exist. We address this issue in line 503f. Consequently, we chose to present the data as boxplots for direct comparison of individual subsections. The boxplot as a form of presentation also has the advantage that some statistical information is directly apparent to the reader, such as the median, the dispersion of the data set, etc.

27) L. 394: Please provide a short paragraph in the introduction how the isotopic signals are interpreted at your site for both d13C (C3/CAM/C4) and dD (amount/source/continentality...). Recent calibration studies provide a nice overview for South Africa. Are there any plants using CAM-metabolism in your studied area?

We added a new section "2 Approach and methodological background". It contains a subsection about "2.3 Plant-wax compound-specific isotopes" (l142ff) which introduces the interpretation of $\delta^{13}$C (l147ff) and δD (l160ff). We also added the requested information on CAM plants (l155f). In the Mkhuze Wetland System no occurrence of CAM plants was reported or observed.

28) L. 413: Please provide a figure showing the *n*-alkane patterns in each sub-environment, as mentioned above.

We chose not to provide a figure showing the respective distribution patterns in each sub-environment (see answer to comment #26).

29) L. 426: Can I see this sub-deviation in any plot?

We changed the sentence accordingly (l571), since the partial deviations mentioned are no longer shown in the figures submitted.

30) L. 430: Please highlight these cluster in the respective figure.

We did as suggested. Figure 6 now displays two different shapes for the swamp sample set to indicate the splitting. We also added the respective information to the figure caption of Figure 6 and gave a reference in the respective discussion part (l576).

31) L. 435: Please refer to the respective figure(s).

We did as suggested (l581).

32) L. 442f.: What is the base of this argumentation? Please introduce to the site-specific dD interpretation earlier.

For the purpose of keeping the internal structure of the section (discussion of bulk parameters first, followed by the relative concentration of *n*-alkanes, the *n*-alkane carbon isotope composition, and finally the *n*-alkane hydrogen isotope composition), we didn't change the order. To provide a more solid argumentation, we reworded the sentence accordingly (l589ff).

33) L. 453: One sentence paragraph which is a result and no discussion. Please modify.

We deleted the sentence.

34) L. 465: […] which use the lake's water as dominant water source? How about the effects of lake water evaporation, salinity to the dD and d$^{13}$C signal, and emergent/submerged plants contributing to the $C_{25}$ to $C_{33}$ *n*-alkane pool?

We rephrased the sentence accordingly (l620f) and added salinity to the lake's characteristics (l619). We additionally, extended and revised the discussion about compound-specific isotopes of plant waxes in the delta/lake area (604ff). A more general explanation on how evaporation affects the isotopic composition is provided in the new section "2 Approach and methodological background" (l157f, l170f).

35) L. 489: Please highlight these clusters in the respective figure. For example, use colored/shaded circles in the background of the data points

We did as suggested (Figure 6).

36) L. 496: Does this statement implies that organic (bio)markers in the wetland are of local origin and thus reflect local eco-hydrological conditions instead of an integrated signal including the wetlands catchment? Please clarify, because it is an essential finding of your study and of great importance for future studies at the site.

No, therefore we clarified the statement (l655ff).

37) L. 517: Remove space before comma.

We did.

38) L. 535: A trap for local organic material – see comment above.

See answer to comment #36.

39) L. 549: Isn't this also the case at your site now – see comments above? Please clarify.

See answer to comment #36.

40) L. 565. Generally, I agree with the conclusions, but doesn't the results demonstrate that the lake sediments are dominated by local organic matter representing very local ecohydrological condition instead of the lakes catchment. This absolutely limits the usage of dD and d$^{13}$C as paleoecological markers because local effects might

overprint the environmental signal, which is a very important finding for future records – although it might be unexpected.

> We are in complete agreement. When interpreting δ¹³C and δD in sediment records derived from downstream areas of such "traps," we would expect to observe the change from an integrated signal to a locally received signal (if the "trap" formed within the time period sampled). This is the reason that the interpretation of sediment records used for paleoecological reconstruction should take into account the geomorphological setting of the sampling area. We added the information on impact for future (paleo)environmental studies in lines 23f and 748f and refined the discussion of subsections (l624f, l655ff).

Figures:

i.  Figure 2: The grey shaded area is very hard to identify. Numbers at the precipitation and evaporation isolines are also very hard to read, please enlarge them.

> We enlarged the numbers from 10pt to 18pt to enhance readability and changes the color of the shaded area for better visibility.

ii.  Figure 3: Please place the figure to the section where it is mentioned in the text. Are there any more recent data than 1996, which is already 25 years ago? Or is this landcover/-usage still present? It seems very important for your study to have the most recent land-cover map for comparison with your data and correct interpretations and implications for future paleo-studies.

> We replaced the figure, although the final placement for publication most likely will be decided by the journal. To the best of our knowledge, this is the most current information on vegetation cover. Data availability for the region is rather sparse.

iii.  Figure 4: Overall, this figure provides a very nice overview of the analyses you did.However, I suggest to remove all the lab-steps, e.g. lipid extraction, and provide a little more details in the respective text and therefore reduce the size of the figure. Just keep the sample (e.g. plant samples [g]), used machines (e.g. GC-FID) and results (e.g. quantity n-alkanes).

> We removed the figure and gave the information as plain text (l262f, l266ff, l304ff).

iv.  Figure 5: Please name the figures in the text first and show them thereafter (Please check for all figures and tables)

> We did.

v.  Figure 8: Why is CS d¹³C of C₂₃ and C₃₅ distinctly more positive compared to the other chain-length? Is there an amount-dependency in the IrMS? Please check to note if you did an amount and/or drift correction of your data for both d¹³C and dD. Additionally, none of the aquatic plant samples show a C₂₅-dominance, which is used as aquatic end-member later, but a distinct C₂₉-dominance which is interpreted as woody end-member later. Moreover, except for 8F, none of the grasses shows a C₃₃-dominance, which is used as grassy end-member in the following. Please clarify these issue, because it is very important for your manuscript.

> There is no amount dependency in the IrMS observable. A specific intra-laboratory cut-off threshold has been established, and only reasonable peaks are integrated. Therefore, we are confident that the more positive values of alkanes C₂₃ and C₃₃ (for Nymphaceae, Figure 8A) and C₃₅ (for Vossia cuspidate, Figure 8E) correspond to reality. The isotopic composition of *n*-alkane C₃₅ (for Cynodon dactylon, Figure 8F) shows no such trend toward higher enriched values compared to the other alkanes, supporting that these values are not artifacts because they were all measured by the same person during the same time period and processed by the same person.

> The explanation on why we chose which *n*-alkanes for what certain vegetation types were refined in the revised subsection "5.1 Plants *n*-alkane distribution patterns as indicators of variable hydrological conditions" (l461ff).

vi. Figure 9: How can $C_{25}$ be the aquatic end-member when your plants show a ($C_{27}$) $C_{29}$ predominance? The same applies for $C_{29}$ and $C_{33}$ for woody and grassy vegetation, respectively. Is there a local study showing that $C_{29}$ is a woody end-member in ZA? There are respective end-member based on you modern aquatic and grassy plant samples, but they are ignored in this figure. Overall, I have to note that the presentation of the data is very nice! However, please consider valid end-member.

We replaced the word "end member" with "marker compound" as mentioned in previous answers. We revised the subsection "5.1 Plants *n*-alkane distribution patterns as indicators of variable hydrological conditions" (l461ff). to provide a better explanation on the choice of the respective marker compounds representing the different clusters.

vii. Figure 10: I really like this figure! Maybe also indicate the pastures, which distinctly contribute to the C4 signal in the floodplains.

We added a fence to the sugar cane plants in the figure to indicate agricultural fields and we are still thankful for your nice compliment.

Tables:

Table 1: I suggest to move this table to the Results section.
We did.

Dear Anonymous Reviewer #2,

we thank you very much for your comments and the time you took to point out potential weaknesses of our manuscript. Below, we will give a point-by-point answer to your comments (all line references refer to line numbers of the marked-up manuscript). An explicit list of all made text changes is provided in the end of the marked-up manuscript version.

Sincerely,
Julia Gensel (on behalf of all co-authors)

**GENERAL**

1. The manuscript is generally very well written and focusses on an interesting and important topic, the characterisation of the organic matter input from Mkhuze Wetland System in South African to Lake St. Lucia. The first problem that struck me when reading the introduction was, that aims and questions are very vague., It is not clear what is meant by assessing the status of the wetland (see below). Is the hydroglogical status meant (drained, undrained) or the soil degradation status or the degradation status of OM? Also using $\delta^{13}C$ and $\delta^2H$ of n-alkanes to characterize sinks and sources is no doubt a fore front method, but for sure not matured enough, to draw conclusions on plant type communities and degradation status of wetlands.

   We thank you for the assessment on our manuscript. We refined the study aims (l52ff). We, in addition, added a new section "2 Approach and methodological background" (l81ff) to give a more detailed introduction to the used methods.

2. As such, compared to such ambitious aims, the introduction is very general. I totally miss discussion of state of the art on $\delta^{13}C$ and $\delta^2H$ organic matter tracing and what it can tell us about sinks and sources. The same holds true for stability of n-alkane concentrations when used as indicators for sediment or organic source attribution.

   We added a new section "2 Approach and methodological background" (l81ff) to give a more detailed introduction to the used methods. The interpretation of *n*-alkane concentration in terms of degradation is used only to support the results from Rock-Eval analyses, which is the main method for determining OM degradation in our study (l528f, l580f).

3. The description of the sampling concept is totally missing. All it states is, that "ten samples where collected." However, Figure 1 displays around 30 sampling sites, so I assume that ten samples for each sub-environment was taken? This is totally unclear. A detailed map of vegetation communities is presented (Figure 3) but it is not at all clear, if all these communities were sampled as possible sources and if so, how many samples, which plant species etc....If the aim is, to track OM in the lake back to these communities, the detailed sampling scheme has to be described.

   The manuscript contains a subsection "3.2 Sampling" (l247ff), which describes, among other things, how many samples were collected and also which plant species were collected. The names of the collected plant species are additionally mentioned in the manuscript in the corresponding results section (subsection "4.3.1 Plant Samples" (l409ff) and the corresponding figure (Figure 7 and its caption). The sentence quoted originally reads "Ten **plant** samples were collected." (l248), while in line 255 the information is given that "a total of 41 surface sediment samples [...] were collected [...]".

We added a reference to Figure 3 to line 251 as the figure is adapted from one of the provided references we based the choice of plant species on. We, in addition, added a sentence regarding the limitation of the size of the sample set (l258f).

4. The results are mainly a listing of all measurements done with differences in numbers and sizes. There is no real information gain for the reader, as none of these results are set into perspective and the discussion does not give a clear link back to these data descriptions. Not even the indices and parameters used are in any way explained in the results section (and only very briefly in the discussion). Variability and differences are hard to assess, as sampling numbers and possible errors are not described. It is not clear if error bars indicated in-field heterogeneity or analytical uncertainty. Figure 8 states that error bars might be intra-laboratory long-term errors.

We agree that we have presented the results in the "Results" section without interpreting them, as it is our understanding that interpretation is part of the "Discussion" section.

Indices and parameters used were explained in the "Material and Methods" section of the manuscript and literature references are provided for all indices and parameters mentioned.

    i. The equations used to calculate the carbon preference index (CPI) and average chain length (ACL) are given in the subsection "3.4 Distributional parameters of n-alkanes" (l344ff).

    ii. The hydrogen index (HI) is introduced in line 290ff in subsubsection "3.3.1 Bulk organic matter analyses".

    iii. R- and I-index are introduced in line 293f within subsubsection "3.3.1 Bulk organic matter analyses".

    iv. The meaning of the indices and parameters is given in the respective parts of the "Discussion" section and, in addition, a link back to the results is provided (also in numbers, if applicable); see e.g. lines 527f, 547ff, 570ff, 597ff. Otherwise, a cross-reference to the respective figure is given (see, e.g., lines 474, 492, 529, 540 etc.).

However, for enhancing the understanding of the used methods we included a new section "2 Approach and methodological background" (l81ff). This section also contains a more detailed introduction and explanation on investigated parameters and indices.

The respective subsections of the "Materials and Methods" section contain information on replicate measurements and sample replicates including standard deviations for every measured $n$-alkane (l253ff, l328f, l338f).

We refined the figure caption of figure 7 concerning the detailed explanation of error bars and their meaning as suggested.

5. The discussion is more a descriptive qualitative narrative of differences found in parameters within and between different sub-ecosystems. Indices for evaluation are not adequately introduced and partly interpreted in a wrong way (e.g., that CN ration of OM would be a general indicator of chemical stability). As such, I can not follow conclusions drawn and can not judge if these conclusions adequately assess the results. One example would be the conclusion "Sedimentary OM in the floodplain and swamp exhibit high variability in their source signatures and degradation status reflecting environmental diversity, with samples from the floodplain characterized by a mixture of degraded OM from the hinterland and fresh OM."(line 575-577). With clearly high ongoing and very variable degradation of OM in these systems, concentrations of organic substances can not be used as a conservative tracer.

To the best of our knowledge, the present study is the first to examine OM properties in the Mkhuze Wetland System. Therefore, no quantitative comparison with previous studies is possible. Intra-system comparison of different sub-areas leads to a qualitative assessment of whether certain characteristics of OM are found to a greater or lesser extent in other sub-areas. We additionally added a new section "2 Approach and methodological background" (l81ff) to give a more detailed introduction to the used methods and interpretation of their results. We also edited the quoted sentence (l738ff).

6. Regarding the isotope tracers used, no un-mixing was done and the values were interpreted in a qualitative way, which is, from my perspective not leading to meaningful conclusions.

We believe that the conclusion we drew are meaningful even though we opted for a qualitative approach. We included a new section "2 Approach and methodological background" (l81ff) and revised the subsection "5.1 Plants *n*-alkane distribution patters as indicators of variable hydrological conditions" to better explain our approach. We deleted the term "end-member" from the respective parts and replaced it with "marker compound", as we think that the former term might have caused confusion regarding our approach.

7. All in all, I would judge this work as containing highly valuable and interesting data and results. But description of sampling concept is inadequate and interpretation of data is qualitive with numerous assumptions I am not sure can be hold.

We thank you for attesting that the data presented are of high interest and value. We believe that we provided an adequate description of the sampling concept in the respective subsection "3.2 Sampling" (247ff). However, we must admit that the usage of the word "These" in a beginning of a sentence, while not referring to the last mentioned object, was very unfortunate and a rather "german grammar" mistake and we are very thankful that you pointed that out.

Overall, we believe that our study contributes to a better understanding of the carbon cycle and carbon storage in the Mkhuze Wetland System, although it was intended to be qualitative. As discussed in detail in the "Discussion" section, our finding that OM is sequestered under current conditions in the swamp area of the wetland system studied may also be found in other wetland systems, suggesting that carbon sequestration in such systems is primarily hydrologically controlled.

**INTRODUCTION**

8. Generally, well written and interesting to read about the Mkhuze Wetland System. However the aim of "assessing the current status" is very vague to me (line 36). Which status do you mean? Hydrological? Soil degradation? Nutrient status?

We revised the respective sentences to better describe the aims of the study (l52ff).

9. I totally miss discussion of state of the art on $\delta^{13}$C and $\delta^2$H organic matter tracing and what it can tell us about sinks and sources. Also, what about the stability of n-alkane concentrations in these systems? Are you sure you can use these as conservative tracers?

We included a new section "2 Approach and methodological background" (l81ff) giving also additional references on recent studies on compound-specific isotope analysis in topsoils (l136, l156, l173). We also added more general information on *n*-alkanes being used as refractory tracers (l130ff), also acknowledging reported limitations (l139). The high CPI values reported corroborate that analyzed *n*-alkanes were not subject to advanced degradation processes.

10. 40-42 this assumes that you have species specific tracers
The sentence was deleted and information is incorporated in the new section "2 Approach and methodological background" (l81ff).

11. 42-45 how can you assess the vegetation type (do you mean plant community?) with $\delta^{13}$C of nalkanes?

The sentence was deleted and information is now incorporated in the new section "2 Approach and methodological background" (l81ff). Introduction to usage of $\delta^{13}$C of *n*-alkanes and interpretation in terms of photosynthetic pathway is introduced from line 147 onwards.

12. 45 hydrological conditions of what? Of the regime under which the plants grew? Of the soil? E.g., wetland, upland, drained? Not sure you can achieve this with $\delta^2$H?

The sentence was deleted and information is now incorporated in the new section "2 Approach and methodological background" (l81ff). Introduction to usage of δD of *n*-alkanes and interpretation in terms of hydrology is provided from line 160 onwards.

13. 51 this is a big aim, to the assess the status of the wetland systems in terms of its filter function and influence on Lake St. Lucia!

The complete paragraph was removed from the introduction. The aims of the study have been revised and are given in line 52ff.

**METHODS**

14. 133 what do you mean by "Ten samples were collected." Of what? Ten repetitions within a site?

The original sentence reads, "Ten plant samples were collected." (l248). This means that we sampled a total of ten plants. The number of replicates of each species is given in lines 252ff. Furthermore, the number of collected surface sediment samples is given in line 255.

15. 137 – 139 these plants were not collected? But all others were? Or these are the ones which you did collect?

We are very thankful for pointing out the wrong usage of the word "These" at the beginning of the sentence and reworded the sentence accordingly (l252).

16. Figure 4 is basically describing standard analysis and could be moved to supporting information.

The Figure has been removed and the sample preparation information is now provided as plain text (l266ff, l304ff).

**RESULTS**

17. 252 what is HI value?

The new section "2 Approach and methodological background" (l81ff) also contains a subsection (2.1, l86ff) about Rock-Eval analyses and the related parameters. Regarding the Hydrogen Index the information is provided in line 91ff.

18. 256 what is R-index?

The new section "2 Approach and methodological background" (l81ff) also contains a subsection ("2.1 Rock-Eval", l86ff) about Rock-Eval analyses and the related parameters. Information on the R-index is provided in line 109f and 115f.

19. 260 what is I index?

The new section "2 Approach and methodological background" (l81ff) also contains a subsection ("2.1 Rock-Eval", l86ff) about Rock-Eval analyses and the related parameters. Information on the I-index is provided in line 111f and 116.

20. For all errors it is not clear from how many reps they are produced, if repetition at all or if this is analytical error. See answer to comment #4.

**DISKUSSION AND CONCLUSIONS**

21. Paragraphs 343 -349 versus 332 – 338: I am not sure I understand you correctly, but this makes not much sense to me. First, you describe the differences in n-alkane patterns within plants, within sites and between different ecosystems, but then you assume that you can take literature values from generally well studied plants, such as trees as source values to be characteristic for your sites?

We revised subsection "5.1 Plants *n*-alkane distribution patters as indicators of variable hydrological conditions" (l461ff) to better explain our approach on identifying marker compounds which are exclusive for certain vegetation types. We also extended the discussion about the $C_{29}$ *n*-alkane (484ff). Although recent studies corroborated its tree-bias and highlighted its ability to act as "responsive sensor" for tropical tree vegetation, we decided to rename the former "woody" cluster to "average" cluster (Figure 8), as the $C_{29}$ *n*-alkane does not fulfill our requirements of exclusive occurrence.

22. The discussion in 4.1. mainly compares the n-alkane concentrations determined in this study in comparison to literature values. But what is the message behind this paragraph?

    We revised the subsection "5.1 Plants *n*-alkane distribution patters as indicators of variable hydrological conditions" (l461ff).

23. Paragraph 4.2.: what is is this telling me regarding the aim of your study, the status of your system, what is the aim of this paragraph? What is the connection to your results?

    We thank you for pointing out that this paragraph (originally intended to provide information applicable to all subsequent subsections (5.2.1 - 5.2.4) caused confusion. We incorporated the information in the new section "2 Approach and methodological background" (l81ff).

24. 369 the C/N ratio is an indicator of chemical stability? I do not think so?

    The complete paragraph has been deleted.

25. Paragraphs 4.2.1. to 4.2.4. describe the variation of the different measured parameters in each of the sub-ecosystem types and tries to induce state of degradation of OM or plant origin. This discussion is qualitative and not really set in perspective to literature values.

    The mentioned subsubsections (now 5.2.1 to 5.2.4) provide the interpretation of the measured variables in terms of OM properties and forms the basis for the following inferences about OM transport pathways within the system. Rock-Eval analyses is fully capable of addressing OM degradation state. Information on this is now provided in the newly included section "2 Approach and methodological background" (l81ff) and its subsection "2.1 Rock-Eval", l86ff.

    As to the best of our knowledge no previous studies regarding OM in the Mkhuze Wetland System and its characteristics exits and therefore no comparison to literature values is possible.

26. Section 4.3. (line 467 – 480) starts with a general description of Mkhuze Wetland System which might be transferred to the methods or the introduction. Or are these statements conclusion from your data? If so, please make the link to your results.

    The first sentence refers to the previous subsection (now 5.2), where interpretations of OM properties from measured parameters are given, and introduces the following conclusion on OM transport pathways. The second sentence extends the interpretation of OM to sediments in general and justifies this with literature references. The following sentences (lines 633ff) are the essence of the interpretation of OM properties and inferred transport pathways. These are related to the available literature and any discrepancies that arise are discussed.

27. 494 – 499 is this general knowledge of literature or is this a conclusion from your results? Please make the link to your data

    We edited the sentence accordingly (l655ff) to highlight results from our study.

28. Some conclusions might be considered speculative.... e.g., from the result that "....the higher hydrogen isotope signature of the sedimentary n-alkanes in the lake probably resulted from a dominant contribution of lakeshore vegetation" (line 464 – 465) the general conclusion is drawn, that "OM in the surface sediments of Lake St. Lucia originates primarily from lakeshore vegetation" (line 501). There is no unmixing of possible source signatures, no quantitative evaluation. This is just one example, which leaves the impression, that conclusions drawn are based on rather qualitative assumptions and might even be speculative.

We revised the respective sentences (l620ff) and the respective subsubsection (l595ff) to enhance the transparency of our conclusions. In combination with the newly included section "2 Approach and methodological background" (l81ff) and its explanation on interpretation of methods used in the study, we are convinced that the reader is not left with the impression, that conclusion drawn are not correct.

---

## Referee Report (RR1)

This is my second review of the study "*Origin, transport, and retention of fluvial sedimentary organic matter in South Africa's largest freshwater wetland, Mkhuze Wetland System*" by Julia Gensel and colleagues. The authors carefully considered the suggestions of my first review. I appreciate the detailed response to each comment and respective modifications in the main text. Overall, the study is sound, but there are still some minor comments to the manuscript. Thus, I suggest *minor revisions* for the study in its current form.

**Specific comments:**

L. 6ff.: It might be good to shortly explain the I-index and R-index in the abstract like you do for leaf wax lipids and their compound-specific isotopic $\delta^{13}C$ and $\delta D$ signature.

L. 98ff.: Please introduce to the ACL as well, which is presented in the results section and table 1. Due to the general nature of this section, please note that both $C_{27}$ and $C_{29}$ are thought to indicate tree-like vegetation while $C_{31}$ and $C_{33}$ are predominantly synthesized by grasses. However, both $C_{29}$ and $C_{31}$ can reflect a mixed signal of trees and grasses. This statement is only given for $C_{31}$ in the introduction while it is described for $C_{29}$ in the discussion section.

L. 124: Maybe modify to […] , i.e., $^{13}C$-enriched *n*-alkanes, […] ?!

L. 144f.: Besides Herrmann et al. (2017, org. geochem.) also Strobel et al. (2020, STOTEN) discuss the effect of evapo(transpi)rative enrichment on the $\delta D$ signature of *n*-alkanes in South Africa. Thus, I suggest to cite both studies here.

L. 337ff.: Is there evidence for dolomite in the catchment/samples which might not be destroyed using HCL without thermal treatment of the samples?

L. 278ff.: Is there any reason why plant samples were treated with a different solvent mixture and additional extracting steps (i.e., MeOH, MeOH:DCM (1:1) and DCM) compared to the sediments (DCM:MeOh 9:1)?

L. 291ff.: How about the recovery of the internal STD (squalane) in the samples and blanks?

**Figure:**

Figure 6: Please provide a legend which enables faster and more intuitive reading of the figure.

Figure 8: To overcome questions of the readership of your MS, I suggest to create box-plots for all chain-length ($C_{23}$ to $C_{35}$) for all sub-environments. Even if you present an extended version of this figure in the supplements would enable the reader to more get a more comprehensive impression of your data. Still, I am a little confused why you present $C_{29}$, which you refer to as mixed signal, while $C_{27}$ and $C_{31}$ might be mixed signals as well. However, the latter two are not presented and you do not present a reason for that.

---

## Author Response (AR2)

**response to the handling associate editor:**

Dear Dr. Sebastian Naeher,

We explicitly thank you for all the work and time you provided to improve our manuscript and to ensure the quality of the review process. In addition, I personally, would like to apologize for the delay of the submission of the revised manuscript and still hope that you are not bothered to much by the situation as well as to thank you for your understanding. Below, we will give a point-by-point answer to your comments (line references given in the answers refer to line numbers of the current marked-up manuscript version).An explicit list of all made text changes is provided in the end of the marked-up manuscript.

Sincerely,

Julia Gensel (on behalf of all co-authors)

1.  Attributing n-C29 alkane to reflect an "average" source does not seem appropriate. If I understand correctly, it appears to be rather undiagnostic indicator in your study, because it is the dominant alkane in diverse plant communities. Is that correct? If so, then I wonder if the term "mixed" would be more appropriate here?
    > We did as suggested and substituted the word "average" with "mixed" within the text and in Figure 8. *The text was changed accordingly.*

2.  Could you please specify the last sentence of the abstract (line 23): "This finding raises important constraints for future environmental studies as the assumption of watershed-integrated signals in sedimentary archives retrieved from downstream lakes or offshore might not hold true in certain settings." I think you would not expect that a lake has the signature from the catchment in all cases, because this will depend on relative contributions of autochthonous vs allochthonous organic matter sources. Catchment signatures can be diluted largely by high productivity and OM export to the sediment. Therefore, you could clarify in the text (e.g. abstract but also elsewhere) that the lake OM signature reflects largely an in-situ signature which seems to largely overprint contributions of allochthonous OM.
    > Often the assumption is made in paleo-environmental studies applying biomarkers that archives retrieved from terminal water bodies, such as lakes or finally the ocean, reflect integrated signals from the whole river catchment. This is the assumption we were investigating, and our results greatly challenge it in settings which are similar to our study site.
    > We generally agree that autochthonous OM sources have the potential to substantially dilute signals retrieved from lake cores regarding certain parameters. Usage of long-chain *n*-alkanes derived from higher plants as biomarkers in combination with bulk analyses by Rock-Eval that clearly proves that OM in lake surface sediment samples do not reflect aquatic autochthonous contributions (line 547ff), show that the signals we

find are predominantly derived from the local surrounding vegetation of the lake, but not from the upstream catchment. *No changes in the text were made.*

3. The last few sentences of Section 2.3 could profit from adding values (or ranges of values) and/or fractionation factors where appropriate, so the reader is made aware of the magnitude of the changes that would be expected. It would be useful to specify "slight dependency" (line 173).

   The section as well as the whole chapter is thought to broaden the potential readership of the manuscript. As it sometimes might be challenging to understand manuscripts when not being totally familiar with the set of applied methods, we decided to give a very general introduction. To maintain the simplicity and comprehensibility to people of other scientific disciplines and focus, we prefer not to include specific values in the whole section. *No changes in the text were made.*

4. When noting I- and R-indices already in the abstract, you should define there what they are.

   We thank you for the suggestion and added the requested information. *The text was changed accordingly.*

5. Line 469 and following (numbers refer to your manuscript with tracked changes) should be reformulated and made more specific, because it may be unclear what you try to say.

   We did as suggested. *The text was changed accordingly.*

6. Line 620: Why is "lakeshore" removed? Isn't this very important here?

   Yes and no. The word "lakeshore" was substituted with "[…], which uses the lake's water as dominant water source.". This fact clearly points towards the shoreline vegetation as source, but by pointing out the usage of the lake water as water source, the phrase is more concise with the reason given (higher hydrogen isotopic signatures). To make it easier for readers without deep knowledge about hydrogen isotopes and their application, we wanted to add this causal relationship. *No changes in the text were made.*

**response to anonymous reviewer #1:**

Dear Anonymous Reviewer #1,

We thank you very much for your second review and positive feedback regarding our applied changes based on the first reviews. Below, we will give a point-by-point answer to your comments (line references given in the answers refer to line numbers of the current marked-up manuscript version).An explicit list of all made text changes is provided in the end of the marked-up manuscript.

Sincerely,

Julia Gensel (on behalf of all co-authors)

**Specific comments**

1.  L. 6ff (now l6f) It might be good to shortly explain the I-index and R-index in the abstract like you do for leaf wax lipids and their compound-specific isotopic $\delta^{13}$C and $\delta$D signature.
    > We did as suggested. *The text was changed accordingly.*
2.  L. 98 ff (now l99ff) Please introduce to the ACL as well, which is presented in the results section and table 1. Due to the general nature of this section, please note that both $C_{27}$ and $C_{29}$ are thought to indicate tree-like vegetation while $C_{31}$ and $C_{33}$ are predominantly synthesized by grasses. However, both $C_{29}$ and $C_{31}$ can reflect a mixed signal of trees and grasses. This statement is only given for $C_{31}$ in the introduction while it is described for $C_{29}$ in the discussion section.
    > We did as suggested. *The text was changed accordingly.*
3.  L. 124 (now l127): Maybe modify to […] , i.e., $^{13}$C-enriched *n*-alkanes, […] ?!
    > We did as suggested. *The text was changed accordingly.*
4.  L. 144f (now l148).: Besides Herrmann et al. (2017, org. geochem.) also Strobel et al. (2020, STOTEN) discuss the effect of evapo(transpi)rative enrichment on the $\delta$D signature of *n*-alkanes in South Africa. Thus, I suggest to cite both studies here.
    > We did as suggested. *The citation was added.*
5.  L. 337ff (now l240ff).: Is there evidence for dolomite in the catchment/samples which might not be destroyed using HCL without thermal treatment of the samples?
    > We discussed that issue thoroughly due to the potential presence of siderite and dolomite. Some organic parameters, such as TOC, were determined by both bulk analyses and Rock-Eval. The results were in good agreement indicating that the HCL treatment worked properly which was corroborated by the yellowish color of the acidic solution. However, parameters which we suspected to potentially be affected by mineral presence weren't incorporated into the manuscript. *No changes in the text were made.*
6.  L. 278ff (now l281ff).: Is there any reason why plant samples were treated with a different solvent mixture and additional extracting steps (i.e., MeOH, MeOH:DCM (1:1) and DCM) compared to the sediments (DCM:MeOh 9:1)?

Yes. Surface sediment samples were extracted using an ASE which enhances the extraction efficiency due to increased temperature and pressure. To obtain a comparable result for plant samples, which are not suitable for the ASE extraction, requires additional solvent use of different polarities to liberate the lipid fractions. *No changes in the text were made.*

7. L. 291ff (now l294ff).: How about the recovery of the internal STD (squalane) in the samples and blanks?

We added the requested information. *The text was changed accordingly.*

**Figure:**

1. Figure 6: Please provide a legend which enables faster and more intuitive reading of the figure.

We did as suggested. *The figure was adjusted accordingly.*

2. Figure 8: To overcome questions of the readership of your MS, I suggest to create box-plots for all chain-length ($C_{23}$ to $C_{35}$) for all sub-environments. Even if you present an extended version of this figure in the supplements would enable the reader to more get a more comprehensive impression of your data. Still, I am a little confused why you present $C_{29}$, which you refer to as mixed signal, while $C_{27}$ and $C_{31}$ might be mixed signals as well. However, the latter two are not presented and you do not present a reason for that.

We performed statistical analyses to identify groups consisting of at least 2 *n*-alkanes (correlation coefficients and results are given in line 408ff). We decided only to show one representative of each identified *n*-alkane group for visualization purposes (l404f). For the interested reader, the complete dataset including all individual *n*-alkane data can be found in the open access data repository Pangaea. *No changes of the figure were made.*

**response to anonymous reviewer #2:**

Dear Anonymous Reviewer #2,

We thank you very much for your second review and positive critique regarding our applied changes based on the first reviews. Below, we will give a point-by-point answer to your comments (line references given in the answers refer to line numbers of the current marked-up manuscript version).An explicit list of all made text changes is provided in the end of the marked-up manuscript.

Sincerely,

Julia Gensel (on behalf of all co-authors)

1. l91 (now l64) do you mean mg H/g?

   The current version "mg HC/g" is correct. HC stands for hydrocarbon which are measured as effluent to determine the hydrogen amount. *No changes in the text were made.*

2. l93 (now l65) "and" missing…

   In the respective line, unfortunately we didn't find the missing "and". *No changes in the text were made.*

3. L95-96 (now l68-69) TOC levels? TOC in %? Fresh plant OM for sure has more than 10 or even 40%?

   This is correct, and we appreciate your awareness very much. *The text was changed accordingly.*

4. L99 (now l71-73) why would aquatic inputs have generally a distinct HI? I assume you want to express, that the HI can sometimes be sued to track sources?

   As organisms contributing to inputs considered as aquatic have a different and distinct composition of the related biomolecules, such as proteins, cellulose etc., they show a distinct HI signature which allows the HI usage as rough source indicator. *No changes in the text were made.*

5. L127 (now l99f) specific origin is relative, e.g., not taxa specific, not even genus specific, please be more precise.....

   We agree that "specific origin" is relative. Our sentence is phrased generally on purpose as indicated by the beginning "In general,[…]". It is thought to give the opening to the subsection and the general concept of biomarkers in contrast to bulk OM methods. We believe that the sentence is phrased in sufficiently general language to not lead the general reader to the conclusion of a more specific source specificity. *No changes in the text were made.*